# Mechanism of membrane pore formation by human gasdermin-D

Estefania Mulvihill[1] [ID], Lorenzo Sborgi[2], Stefania A Mari[1], Moritz Pfreundschuh[1], Sebastian Hiller[2] [ID] &
Daniel J Müller[1,*] [ID]

## Abstract

Gasdermin-D (GSDMD), a member of the gasdermin protein family, mediates pyroptosis in human and murine cells. Cleaved by inflammatory caspases, GSDMD inserts its N-terminal domain (GSDMD[Nterm]) into cellular membranes and assembles large oligomeric complexes permeabilizing the membrane. So far, the mechanisms of GSDMD[Nterm] insertion, oligomerization, and pore formation are poorly understood. Here, we apply high-resolution (≤ 2 nm) atomic force microscopy (AFM) to describe how GSDMD[Nterm] inserts and assembles in membranes. We observe GSDMD[Nterm] inserting into a variety of lipid compositions, among which phosphatidylinositide (PI(4,5)P2) increases and cholesterol reduces insertion. Once inserted, GSDMD[Nterm] assembles arc-, slit-, and ring-shaped oligomers, each of which being able to form transmembrane pores. This assembly and pore formation process is independent on whether GSDMD has been cleaved by caspase-1, caspase-4, or caspase-5. Using time-lapse AFM, we monitor how GSDMD[Nterm] assembles into arc-shaped oligomers that can transform into larger slit-shaped and finally into stable ring-shaped oligomers. Our observations translate into a mechanistic model of GSDMD[Nterm] transmembrane pore assembly, which is likely shared within the gasdermin protein family.

Keywords cell death; inflammation; gasdermin-D pore assembly; time-lapse high-resolution atomic force microscopy; transmission electron microscopy
Subject Categories Autophagy & Cell Death; Immunology
The EMBO Journal (2018) 37: e98321

## Introduction

Pyroptosis is an inflammatory type of programmed cell death that is triggered by a variety of threats, including intracellular pathogen- or host-derived perturbations of the cytosol (Bergsbaken *et al*, 2009; Jorgensen & Miao, 2015). Pyroptosis is characterized by cell swelling, cell membrane disruption, and release of cytoplasmatic contents including pro-inflammatory molecules, such as the matured form of interleukin-1β and interleukin-18. The pro-inflammatory molecules then recruit immune cells to the active infection site to restrict replication of intracellular pathogens (Miao *et al*, 2010). However, while pyroptosis protects multicellular organisms from invading pathogens, an excessive activation of this pathway can also lead to tissue damage, organ failure, and lethal septic shock (Aziz *et al*, 2013, 2014). Pyroptosis is characterized by its dependence on a dedicated set of cysteine-dependent proteases, the so-called inflammatory caspases, which include caspase-1, caspase-4, and caspase-5 in humans and caspase-1 and caspase-11 in mice (Cookson & Brennan, 2001; Aachoui *et al*, 2013). These caspases are activated by inflammasomes, multi-protein complexes that assemble upon recognizing certain cytosolic danger- or pathogen-associated molecular patterns by cytosolic receptors (Broz & Dixit, 2016). Whereas the basis of inflammasome assembly and caspase activation has been well resolved (Lamkanfi & Dixit, 2014), the molecular mechanism causing pyroptosis remained unclear for over a decade (Rühl & Broz, 2016). In 2015, three landmark studies identified gasdermin-D (GSDMD) as an essential mediator of pyroptosis in human and murine cells (He *et al*, 2015; Kayagaki *et al*, 2015; Shi *et al*, 2015).

Gasdermin-D is a human member of the gasdermin (GSDM) protein family, which is processed by inflammatory caspases and cleaved into an N-terminal (GSDMD[Nterm]) and a C-terminal (GSDMD[Cterm]) fragment. The N terminus by itself can induce pyroptosis when expressed ectopically, and the overexpression of GSDMD[Cterm] is found to block GSDMD[Nterm]-dependent pyroptosis (He *et al*, 2015; Kayagaki *et al*, 2015; Shi *et al*, 2015). Recently, it has been shown that GSDMD[Nterm] is cytotoxic because it can target, insert, and permeabilize cellular membranes by assembling transmembrane pores (Aglietti *et al*, 2016; Ding *et al*, 2016; Liu *et al*, 2016; Sborgi *et al*, 2016). Therefore, GSDMD[Nterm] represents a novel class of pore-forming protein. Furthermore, it has been shown that other members of the gasdermin protein family, gasdermin-A (GSDMA), gasdermin-A3 (GSDMA3), and gasdermin-E (GSDME), form transmembrane pores in lipid membranes (Ding *et al*, 2016; Rogers *et al*, 2017; Wang *et al*, 2017). Since all gasdermin family members share similar N-terminal domains, it has been proposed

1  Department of Biosystems Science and Engineering, Eidgenössische Technische Hochschule (ETH) Zurich, Basel, Switzerland
2  Biozentrum, University of Basel, Basel, Switzerland
   *Corresponding author. Tel: +41 6136 73307; E-mail: daniel.mueller@bsse.ethz.ch

that the members exhibit comparable pore-forming activities and share common mechanisms of pore formation (Shi *et al*, 2015; Ding *et al*, 2016; Aglietti & Dueber, 2017; Rogers *et al*, 2017). However, the mechanism by which GSDMD[Nterm] and generally the gasdermin protein family members oligomerize and assemble lytic pores remains largely unclear.

Here, we apply high-resolution (≤ 2 nm) atomic force microscopy (AFM) to characterize the oligomers formed by GSDMD[Nterm] cleaved from GSDMD by human inflammatory caspase-1, caspase-4, or caspase-5 in lipid membranes. To further understand the role of lipids, we characterize GSDMD[Nterm] binding, insertion, and oligomerization to eleven different lipid compositions. Using time-lapse AFM, we image at high temporal and spatial resolution the mechanisms by which GSDMD[Nterm] assembles and forms transmembrane pores. We observe that GSDMD[Nterm] assembles into arc- and slit-shaped oligomers, which form dynamic structures that transform into larger thermodynamically stable ring-shaped oligomers. The insights highlight the mechanism by which GSDMD[Nterm] self-assembles and forms transmembrane pores.

## Results

### GSDMD[Nterm] oligomerization in lipid membranes

Recently, it has been shown that GSDMD[Nterm] binds and inserts into lipid membranes where it can assemble arc-, slit-, and ring-shaped oligomers, some of which being able to form transmembrane pores (Aglietti *et al*, 2016; Ding *et al*, 2016; Liu *et al*, 2016; Sborgi *et al*, 2016). It has been also reported that the lipid composition of the membrane directly influences whether GSDMD can permeabilize liposomes (Aglietti *et al*, 2016; Ding *et al*, 2016; Liu *et al*, 2016). To link both observations, we wanted to image GSDMD[Nterm] oligomers formed in membranes composed of different lipids using high-resolution AFM at physiologically relevant conditions. We thus prepared lipid membranes assembled from a mixture of phosphatidylcholine (POPC), dioleoylphosphatidylglycerol (DOPG), dioleoylphosphatidylserine (DOPS), dioleoylphosphatidylethanolamine (DOPE), and cardiolipin (CL) (40:20:10:20:10 molar ratio) or from *Escherichia coli* polar lipid extract without and with cholesterol being added. The lipid membranes were supported by freshly cleaved atomically flat muscovite mica in buffer solution at 37°C (Materials and Methods). Each supported lipid membrane (SLM) was imaged by AFM to control whether it uniformly covered the mica support and showed no defects such as holes, cracks, or stacked lipid membranes (Appendix Fig S1). Afterwards, we incubated the SLM with 0.5 μM GSDMD and 0.1 μM caspase-1 for 60 min at 37°C, which resulted in the formation of cytotoxic GSDMD[Nterm] and of GSDMD[Cterm] (Appendix Fig S2), and imaged the sample by force–distance curve-based AFM (FD-based AFM; Dufrene *et al*, 2013, 2017) in buffer solution (Fig 1). In the past few years, FD-based AFM has proven to record high-resolution topographs (≤ 2 nm) of membrane proteins, including pore-forming proteins and toxins (Medalsy *et al*, 2011; Leung *et al*, 2014; Pfreundschuh *et al*, 2014; Mulvihill *et al*, 2015; van Pee *et al*, 2016). The nanoscopic approach was recently developed towards reaching an exceptional sensitivity at sub-nanometer resolution, which allows, for example, the imaging of single

polypeptide loops of native membrane proteins in the unperturbed conformation (Medalsy *et al*, 2011; Pfreundschuh *et al*, 2013, 2014).

Atomic force microscopy topographs recorded of GSDMD[Nterm] incubated to SLMs made from POPC, DOPG, DOPS, DOPE, and CL (40:20:10:20:10 molar ratio) showed oligomers having slit-, ring-, and rarely arc-shaped structures (Fig 1A, D–F). The height analysis showed that the slit-shaped GSDMD[Nterm] oligomers protruded 3.6 ± 0.2 nm (mean ± SD; $n = 74$) from the lipid surface and the ring-shaped protruded 3.6 ± 0.3 nm ($n = 277$; Fig 1G and M, and Appendix Table S1). The analysis also showed that each of the oligomeric structures occasionally formed pores penetrating through the lipid membrane. The diameter of the ring-shaped oligomers distributed widely from 13.5 to 33.5 nm showing a mean of 22.6 ± 0.3 nm (mean ± SE; $n = 288$; Fig 1J). Taken together, the relatively low height of the GSDMD[Nterm] oligomers protruding from the lipid membrane and the fact that the oligomers form transmembrane pores suggest that they inserted into the membrane. The observation that arc-, slit-, or ring-shaped oligomers of self-inserting proteins can form transmembrane pores is similar to what has been described earlier for other pore-forming proteins including listeriolysin O (Mulvihill *et al*, 2015; Podobnik *et al*, 2015), perfringolysin O (Czajkowsky *et al*, 2004), pneumolysin (Sonnen *et al*, 2014; van Pee *et al*, 2016), streptolysin O (Palmer *et al*, 1998), suilysin (Leung *et al*, 2014), Bax (Salvador-Gallego *et al*, 2016), and perforin (Metkar *et al*, 2015). Furthermore, the relatively wide distribution of the diameter of ring-shaped oligomers and the coexistence of arc-, slit-, and ring-shaped oligomers indicates that the assembly of transmembrane pores by GSDMD[Nterm] is a structurally rather flexible process.

Incubated to SLMs made from *E. coli* polar lipid extract, GSDMD[Nterm] also formed slit-, ring-, and rarely arc-shaped oligomers (Fig 1B, D–F). The slit- and ring-shaped GSDMD[Nterm] oligomers protruded 3.5 ± 0.3 nm (mean ± SD; $n = 135$) and 3.5 ± 0.3 nm ($n = 164$) from the lipid surface, respectively (Fig 1H and M, and Appendix Table S1). The height analysis also showed that the oligomeric structures occasionally formed transmembrane pores regardless of their shape. The diameter of the ring-shaped oligomers distributed from 15.0 to 33.4 nm showing a mean of 23.1 ± 0.7 nm (mean ± SE; $n = 183$) (Fig 1K).

To investigate the effect of cholesterol on the binding, oligomeric assembly, and pore formation of GSDMD[Nterm], we incubated GSDMD[Nterm] on SLMs made from *E. coli* polar lipid extract supplemented with cholesterol (70:30 weight ratio). The AFM topographs showed that GSDMD[Nterm] again assembled arc-, slit-, and ring-shaped oligomers (Fig 1C). The slit- and ring-like oligomers protruded 3.5 ± 0.3 nm (mean ± SD; $n = 35$) and 3.4 ± 0.4 nm ($n = 63$) from the surface of the lipid membrane, respectively (Fig 1I and M, and Appendix Table S1). The height analysis also showed that the oligomeric structures occasionally formed transmembrane pores. Again, the diameter of the ring-shaped oligomers widely ranged from 13.5 to 38.0 nm with a mean of 21.2 ± 0.4 nm (mean ± SE; $n = 94$; Fig 1L). However, the probability to find GSDMD[Nterm] oligomers in cholesterol supplemented SLMs was reduced by 85% (43 ± 11 oligomers per μm²; mean ± SE) compared to finding oligomers in SLMs made from only *E. coli* polar lipid extract (274 ± 40 oligomers per μm²). This result shows that cholesterol hindered the binding of GSDMD[Nterm] to the lipid

**Figure 1. Characterization of the assembly of GSDMD^Nterm oligomers.**

A–C   AFM topographs showing GSDMD^Nterm oligomers formed on supported lipid membranes (SLMs) assembled from (A) POPC, DOPG, DOPS, DOPE, and CL (40:20:10:20:10 molar ratio), (B) *E. coli* polar lipid extract, or (C) *E. coli* polar lipid extract and cholesterol (70:30 weight ratio). Scale bars, 50 nm.

D–F   Topographs showing (D) arc-like, (E) slit-like, and (F) ring-like GSDMD^Nterm oligomers formed on SLMs assembled from POPC, DOPG, DOPS, DOPE, and CL (40:20:10:20:10 molar ratio), *E. coli* polar lipid extract, or *E. coli* polar lipid extract and cholesterol. Scale bars, 10 nm.

G–I   Height profiles of GSDMD^Nterm oligomers measured along the red lines in the AFM topographs (A-C). Black dashed lines indicate the SLM surface (0 nm height). Scale bars, 10 nm.

J–L   Distribution of GSDMD^Nterm ring diameters formed into SLMs made from (J) POPC, DOPG, DOPS, DOPE, and CL, (K) *E. coli* polar lipid extract and (L) *E. coli* polar lipid extract and cholesterol. Black lines are Gaussian fits determining the mean ± SE values given for each distribution.

M   Average height of GSDMD^Nterm oligomers protruding from the lipid bilayer. Heights are measured for ring- (black) and slit-shaped (gray) oligomers formed in SLMs made from (i) POPC, DOPG, DOPS, DOPE, and CL, (ii) *E. coli* polar lipid extract or (iii) *E. coli* polar lipid extract and cholesterol. Bars represent average and error bars SD.

Data information: The full color range of the topographs corresponds to a vertical scale of 12 nm. Averages and errors are summarized in Appendix Table S1.

membrane. However, it also suggests that once GSDMD^Nterm bound and inserted into the membrane, cholesterol does not hinder GSDMD^Nterm to self-assemble oligomeric transmembrane pores.

## GSDMD^Nterm oligomerization in liposomes

We have observed that GSDMD cleaved by caspase-1 oligomerizes on supported lipid membranes. Next, we wanted to address whether GSDMD^Nterm also inserts slit-, arc-, and ring-like oligomers in non-supported liposomes. To answer this question, we made SLMs from *E. coli* polar lipid extract, which were incubated with GSDMD at 37°C without caspase and imaged the sample by AFM (Fig EV1). Simultaneously, we prepared liposomes from *E. coli* polar lipid extract, which, suspended in buffer solution, were incubated overnight with GSDMD at 37°C without caspase and then imaged by negative-stain transmission electron

microscopy (TEM; Fig EV2). None of the approaches showed GSDMD oligomers in either liposomes or supported membranes. Next, we incubated GSDMD with caspase-1 for up to 12 h (overnight) at 37°C and imaged the samples by AFM and TEM (Figs EV1 and EV2). The samples showed no GSDMD[Nterm] oligomers in the absence of lipids. Next, we incubated the liposomes overnight at 37°C with GSDMD and caspase-1 and imaged the sample by TEM (Fig EV3). Yet, the liposomes were covered with arc-, slit-, and ring-like GSDMD[Nterm] oligomers. These controls, which confirm previous experiments (Ding *et al*, 2016; Sborgi *et al*, 2016), show that GSDMD alone cannot bind to lipid membranes and assemble oligomers. However, they show in addition that GSDMD[Nterm] requires lipid membranes to form oligomers and that the formation of arc-, slit-, and ring-like oligomers is not restricted to SLMs.

## Opposing roles of phosphoinositide and cholesterol

Phosphoinositides and cholesterol, which are both present in mammalian cell membranes, have been proposed to play a role in mediating the interaction of GSDMD[Nterm] with cellular membranes (Ding *et al*, 2016; Aglietti & Dueber, 2017). Whereas phosphoinositides have been shown to facilitate GSDMD[Nterm] binding (Ding *et al*, 2016), cholesterol has been suggested to reduce GSDMD[Nterm] pore formation (Ding *et al*, 2016; Sborgi *et al*, 2016), such as confirmed by our experiments above (Fig 1). We thus wanted to investigate whether the presence of phosphoinositide not only facilitates the binding of GSDMD[Nterm] to lipid membranes but also changes its oligomerization. Furthermore, we wanted to test whether cholesterol also affects the binding of GSDMD[Nterm] to phosphoinositide-containing membranes.

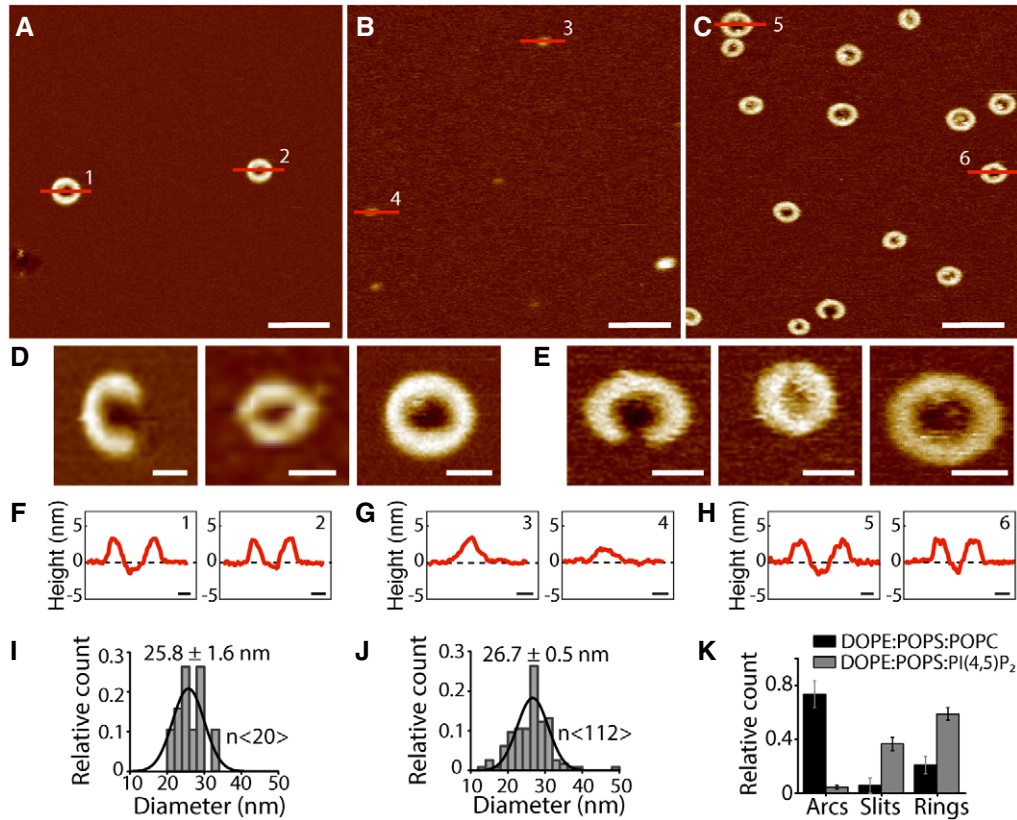

**Figure 2. Effect of phosphoinositide on the assembly of GSDMD[Nterm] oligomers and pores.**

A–C  AFM topographs showing GSDMD[Nterm] oligomers formed on SLMs made from (A) POPS, DOPE, and POPC (35:25:40 molar ratio); (B) POPS, DOPE, and POPI (35:25:40 molar ratio); and (C) POPS, DOPE, and PI(4,5)P2 (35:25:40 molar ratio).

D, E  Topographs showing arc-, slit-, and ring-like GSDMD[Nterm] oligomers formed on SLMs assembled from (D) POPS, DOPE, and POPC (35:25:40 molar ratio) and (E) POPS, DOPE, and PI(4,5)P2 (35:25:40 molar ratio).

F–H  Height profiles of GSDMD[Nterm] oligomers measured along the red lines indicated in the topographs (A–C). Black dashed lines indicate the SLM surface (0 nm height).

I–J  Diameter of GSDMD[Nterm] ring-shaped oligomers formed on SLMs made from (I) POPS, DOPE, and POPC and (J) POPS, DOPE, and PI(4,5)P2. Black lines are Gaussian fits determining the mean ± SE values given for each distribution.

K  Number of arc-, slit-, and ring-shaped GSDMD[Nterm] oligomers formed on SLMs made from POPS, DOPE, and POPC (black bars) or POPS, DOPE, and PI(4,5)P2 (gray bars). Bars present averages and error bars SD.

Data information: The full color range of the topographs corresponds to a vertical scale of 12 nm. Scale bars, 100 nm (A–C), 20 nm (D, E), and 10 nm (F–H). Averages and errors are given in the text and summarized in Appendix Table S1.

First, we addressed the role of phosphoinositides in the binding and assembly of GSDMD$^{Nterm}$ (Fig 2). We thus assembled SLMS from phosphatidylserine (POPS), DOPE, and POPC (35:25:40 molar ratio), from POPS, DOPE, and phosphatidylinositol (POPI) (35:25:40 molar ratio), and from POPS, DOPE, and phosphatidylinositide (PI (4,5)P2) (35:25:40 molar ratio). Each of the SLMs was incubated with 0.5 μM GSDMD and 0.1 μM caspase-1 for 60 min at 37°C, as described above. The AFM topographs showed that GSDMD$^{Nterm}$ binds to SLMs made from POPS, DOPE, and POPC, where it assembled arc-, slit-, and ring-like oligomers (Fig 2A and D). Each of the oligomers could form transmembrane pores (Fig 2F). However, we did not observe GSDMD$^{Nterm}$ binding to SLMs made from POPS, DOPE, and POPI (Fig 2B and G). This effect is expected since phosphatidylinositol lacks the phosphate groups of phosphoinositides, which facilitate GSDMD$^{Nterm}$ binding (Ding *et al*, 2016). In stark contrast, however, GSDMD$^{Nterm}$ bound at considerably higher frequency to SLMs made from POPS, DOPE, and PI(4,5)P2 (Fig 2C, E and H) than to SLMs made from POPS, DOPE, and POPC (Fig 2A). The data also showed that the presence of PI(4,5)P2 had no influence on the diameter of ring-like oligomers (Fig 2I–J), while the occurrence of arc-like oligomers reduced and that of slit- and ring-like oligomers increased (Fig 2K). The results thus show that PI(4,5)

P2 enhances the binding and oligomerization of GSDMD$^{Nterm}$ and enhanced the occurrence of slit- and ring-like oligomers.

Next, we investigated the effect of cholesterol in phosphoinositide-containing membranes on GSDMD$^{Nterm}$ binding and assembly. Therefore, we assembled SLMs from POPS, POPC, DOPE, and PI (4,5)P2 (35:30:25:10 molar ratio) and from the same lipid mixture supplemented with 15 and 30% cholesterol (molar ratio). Each of the SLMs was incubated with GSDMD and caspase-1 as described above. The AFM topographs showed that in the presence of 15% cholesterol less GSDMD$^{Nterm}$ bound to SLMs (Fig EV4). Finally, in the presence of 30% cholesterol, the binding of GSDMD$^{Nterm}$ was largely suppressed. In summary, the experiments showed cholesterol to reduce the binding of GSDMD$^{Nterm}$ to lipid membranes both in the absence (Fig 1C) and in the presence (Fig EV4) of phosphoinositide.

### GSDMD cleaved by different caspases forms similar oligomers

Gasdermin-D can be cleaved by different inflammatory caspases, including human caspase-1, caspase-4, and caspase-5, and murine caspase-1 and caspase-11 (Kayagaki *et al*, 2015; Shi *et al*, 2015). Thereby, the caspases are activated by different pathways (Kayagaki *et al*, 2015; Shi *et al*, 2015; Vigano *et al*, 2015; Aglietti *et al*, 2016).

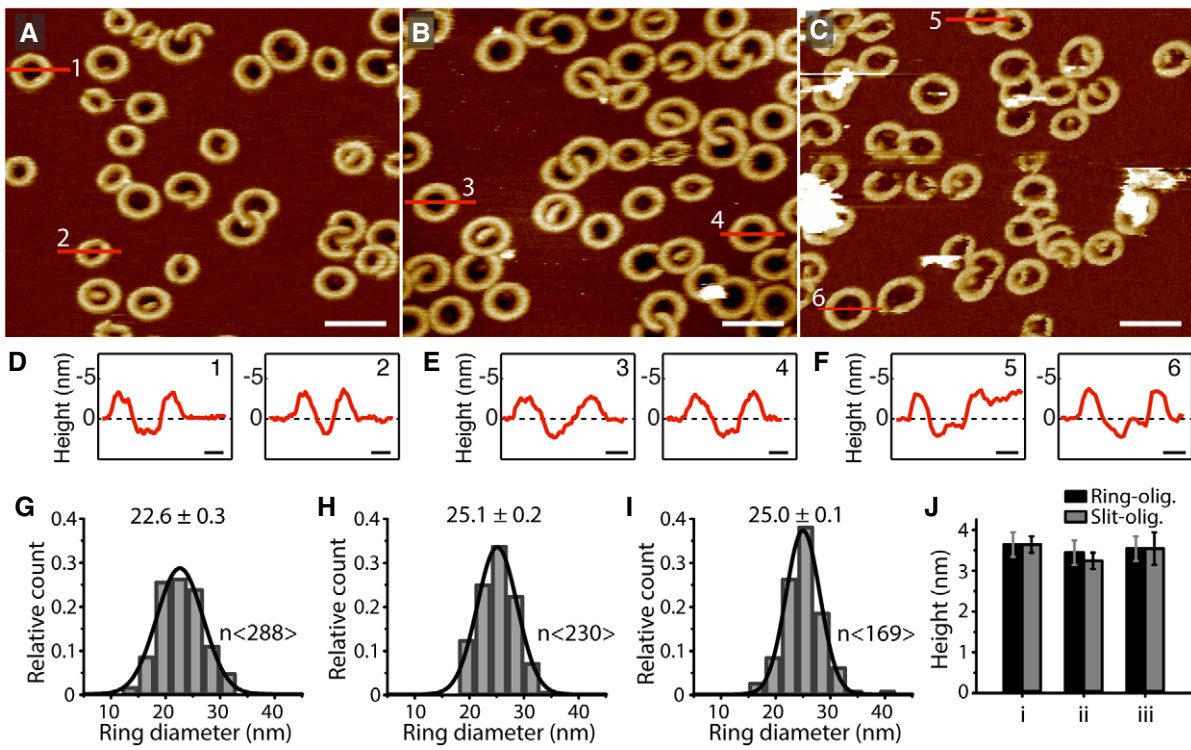

**Figure 3.** **Characterization of GSDMD$^{Nterm}$ oligomers after cleavage of GSDMD by either caspase-1, caspase-4, or caspase-5.**

A–C AFM topographs of POPC, DOPG, DOPS, DOPE, and CL (40:20:10:20:10 molar ratio) SLMs incubated with (A) GSDMD and caspase-1, (B) GSDMD and caspase-4 or (C) GSDMD and caspase-5. The full color range of AFM topographs corresponds to a vertical scale of 12 nm. Scale bars, 50 nm.

D–F Height profiles of GSDMD$^{Nterm}$ oligomers measured along the red lines in the AFM topographs (A–C). Black dashed lines indicate the SLM surface (0 nm height). Scale bars, 10 nm.

G–I Distribution of the diameter of ring-shaped oligomers formed by GSDMD cleaved by (G) caspase-1, (H) caspase-4, or (I) caspase-5. Black lines are Gaussian fits determining the mean ± SE values given for each distribution. The distribution (G), which was taken from Fig 1J, is again shown to allow a better comparison.

J Average height of GSDMD$^{Nterm}$ oligomers protruding from the lipid bilayer. Shown are heights measured for ring- (black) and slit-shaped (gray) oligomers cleaved by (i) caspase-1, (ii) caspase-4, or (iii) caspase-5. Bars present averages and error bars SD. Averages and errors are summarized in Appendix Table S2.

To investigate whether the caspase type affects the membrane binding, assembly into oligomers, and pore formation of the cleavage product GSDMD$^{Nterm}$, we incubated SLMs made from POPC, DOPG, DOPS, DOPE, and CL (40:20:10:20:10 molar ratio) with GSDMD in the presence of either caspase-1, caspase-4, or caspase-5 at 37°C. Control experiments showed that each of the caspases cleaved GSDMD into the N-terminal fragment GSDMD$^{Nterm}$ (Appendix Fig S2), known to induce pyroptosis (He *et al*, 2015; Kayagaki *et al*, 2015; Shi *et al*, 2015; Aglietti *et al*, 2016; Ding *et al*, 2016; Liu *et al*, 2016; Sborgi *et al*, 2016). After an incubation time of 60 min, the sample was rinsed to remove GSDMD and caspase, and imaged by FD-based AFM in buffer solution at room temperature (Fig 3 and

Appendix Table S2). Regardless which of the three caspases cleaved GSDMD, we observed GSDMD$^{Nterm}$ oligomers having no considerable differences in terms of their shape (Fig 3A–C), height protruding above the lipid surface and capability of forming transmembrane pores (Fig 3D–F and J). Also, the diameter of the ring-shaped oligomers did not change considerably (Fig 3G–I). Furthermore, the amount of GSDMD$^{Nterm}$ oligomers bound to the membrane was largely independent of which inflammatory caspase was used. The results show that no matter which caspase cleaves GSDMD during incubation, the cleavage product GSDMD$^{Nterm}$ displays very similar properties of binding and inserting into the lipid membrane and of subsequent oligomerization and pore formation. GSDMD can thus

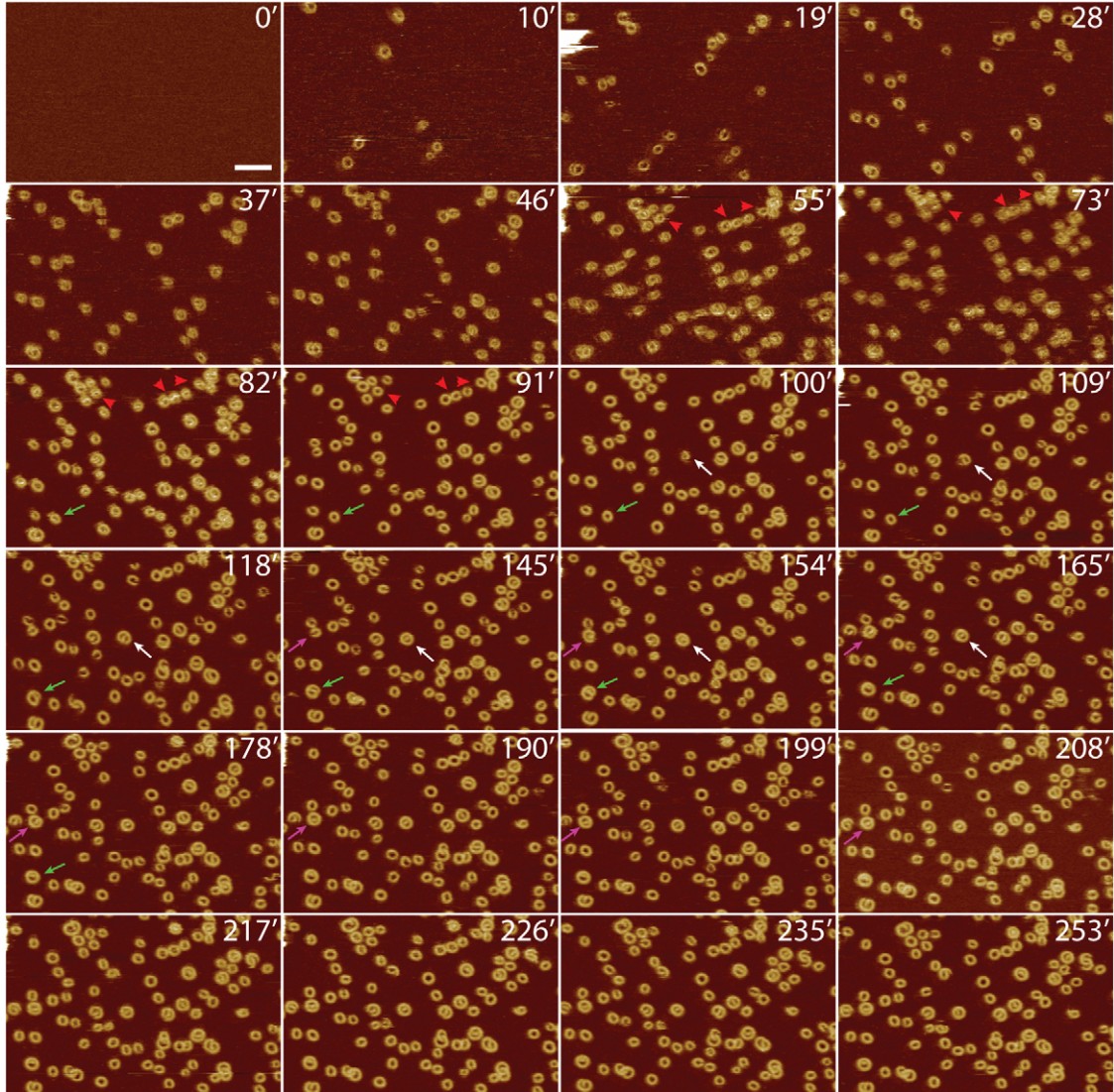

**Figure 4. Time-lapse topographs showing GSDMD$^{Nterm}$ oligomerization and pore formation.**

The first FD-based AFM topograph of a SLM made from POPC, DOPG, DOPS, DOPE, and CL (40:20:10:20:10 molar ratio) was used as a control to show that the lipid membrane was defect-free. The defect-free SLM was then incubated with GSDMD and caspase-1 in buffer solution at 37°C. Recorded at different time points of the incubation (indicated by time stamps in min), the time-lapse AFM topographs follow the progress of GSDMD$^{Nterm}$ binding and assembly to the SLM. Arrows indicate examples of GSDMD$^{Nterm}$ slit-shaped oligomers that through the addition of GSDMD$^{Nterm}$ grow into larger ring-shaped oligomers. Red arrowheads indicate remnants inside oligomers, which disappear with time. Time-lapse FD-based AFM topographs were recorded in buffer solution at 37°C as described (Materials and Methods). The full color range of the topographs corresponds to a vertical scale of 10 nm. Scale bar, 50 nm.

integrate the input from different signaling pathways into the formation of lytic membrane pores.

## Imaging GSDMD^Nterm oligomerization and pore formation

To study the process of GSDMD$^{Nterm}$ oligomerization and pore formation in a time-dependent manner, we conducted time-lapse AFM of SLMs made from POPC, DOPG, DOPS, DOPE, and CL (40:20:10:20:10 molar ratio) while incubating them with GSDMD and caspase-1 at 37°C. Before of the incubation, we imaged the SLM to confirm its structural integrity and the absence of membrane defects (Fig 4 and Appendix Fig S3 and S4). Complete and defect-free SLMs were then incubated with GSDMD and caspase-1. After 10 min of incubation, we observed GSDMD$^{Nterm}$ to form slit-, arc-, and ring-shaped structures on the SLM. Over time, arc- and slit-like oligomers grew from one or both ends by the addition of GSDMD$^{Nterm}$ monomers or oligomers. Finally, the growing ends of arc- and/or slit-shaped oligomers fused with each other to form larger ring-shaped oligomers (Fig 4, Appendix Fig S3 and S4, arrows). Occasionally, the ring-shaped GSDMD$^{Nterm}$ pores showed remnants inside the transmembrane pore, which may have presented fragments from the fusing oligomers and lipid membrane. As such remnants could disappear with time, without the oligomer changing in size, we assume that they exited from the pore into the solution (Fig 4, red arrow heads). During the fusion of oligomers, the surface area of the pore steadily increased until the fused

oligomer reached its final size (Fig EV5). Analysis of the time-lapse AFM topographs revealed that $4.7 \pm 2.5$ % (mean $\pm$ SE, $n_{Experiments} = 3$, $n_{Oligomers} = 2,718$) of the ring-like oligomers assembled from arc-like oligomers and $79.5 \pm 9.8$ % from slit-like oligomers. $24.8 \pm 7.4$ % of the GSDMD$^{Nterm}$ oligomers were already observed as rings. It may be assumed that the already-assembled GSDMD$^{Nterm}$ rings formed faster than the time resolution of the time-lapse topographs ($\approx$ 5–12 min, see Fig 4, Appendix Fig S3 and S4). Throughout the time-lapse experiments, the number of ring-shaped GSDMD$^{Nterm}$ oligomers increased while the number of arc-like oligomers remained at low levels and that of slit-like oligomers reduced to lower levels (Fig EV6). This indicates that arc- and slit-like oligomers, which continuously inserted from the buffer solution into the membrane, represent intermediate states in the assembly pathway that subsequently fused them into more stable and larger ring-like GSDMD$^{Nterm}$ oligomers. Because the fusion of arc- and slit-like oligomers into ring-like oligomers was observed in SLMs made from a large variety of different lipid mixtures, we wanted to test whether this dynamic fusion process is also observed in a single lipid type. Indeed, inserting in SLMs assembled from pure POPG or pure POPC, GSDMD$^{Nterm}$ showed similar details of the assembly and fusion process of arc-, slit-, and ring-like GSDMD$^{Nterm}$ oligomers (Appendix Fig S5). Overall these results indicate that mechanistic steps forming the pathway of GSDMD$^{Nterm}$ oligomerization and pore formation, but not necessarily their kinetic parameters, are highly robust in different lipid environments.

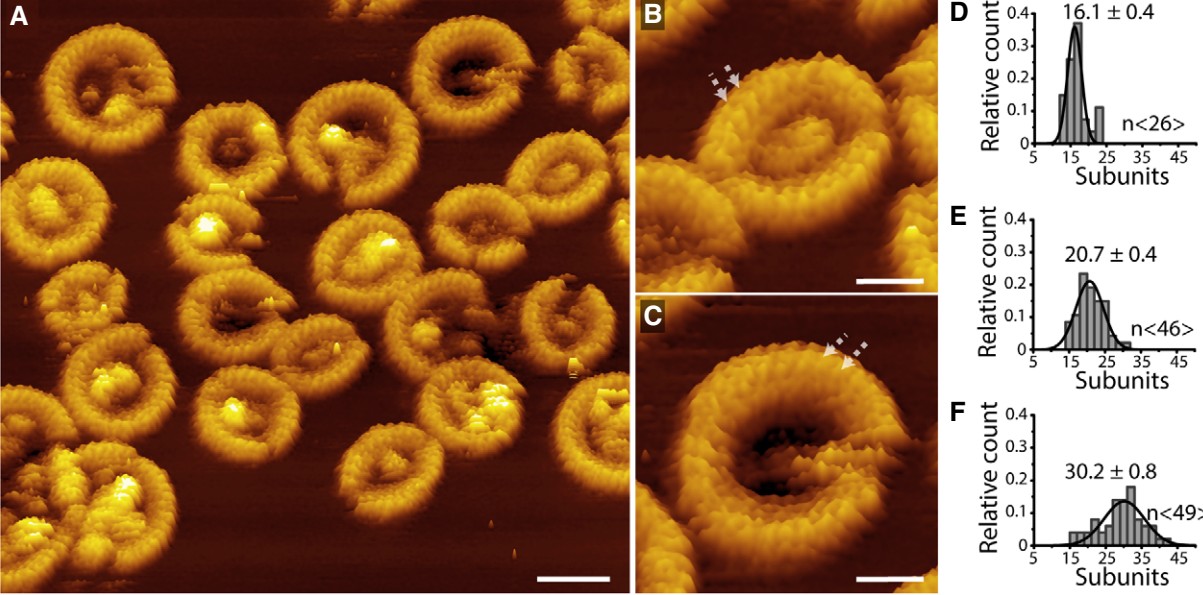

**Figure 5.  Arc-, slit-, and ring-shaped GSDMD^Nterm oligomers imaged at sub-nanometer resolution.**

A    High-resolution AFM topograph of GSDMD$^{Nterm}$ oligomers assembled on SLMs made from POPC, DOPG, DOPS, DOPE, and CL (40:20:10:20:10 molar ratio). After SLMs have been incubated with GSDMD and caspase-1 for 60 min at 37°C, FD-based AFM topograph was recorded in buffer solution at room temperature (Materials and Methods).

B, C    GSDMD$^{Nterm}$ oligomers shown at higher magnification. Arrows indicate adjacent subunits of oligomers, which show an average distance of $2.3 \pm 0.3$ nm (mean $\pm$ SD, $n = 117$).

D–F    Average number of subunits per (D) arc-, (E) slit-, and (F) ring-shaped GSDMD$^{Nterm}$ oligomer. Black lines are Gaussian fits used to determine mean $\pm$ SE of the data, and $n$ indicates the number of oligomers measured.

Data information: The full color range of the topographs corresponds to a vertical scale of 10 nm. Scale bars, 20 nm (A) and 10 nm (B, C).

## GSDMD^Nterm oligomers grow pores without vertical collapse

Pore growing mechanisms without the vertical collapse of pore-forming proteins have been described for members of the membrane attack complex/perforin (MACPF) family (Lukoyanova et al, 2015; Metkar et al, 2015; Leung et al, 2017), whereas a prepore-to-pore transitions associated with a vertical collapse have been reported for cholesterol-dependent cytolysins (CDCs; Czajkowsky et al, 2004; Tilley et al, 2005; Leung et al, 2014; van Pee et al, 2016). To see whether a vertical collapse is associated with the assembly and pore-forming of GSDMD^Nterm oligomers, we analyzed the time-lapse AFM topographs for the heights of the oligomers protruding from the surface of the lipid membrane. During the entire course of the time-lapse experiments (Fig 4, Appendix Fig S3 and S4), the GSDMD^Nterm oligomers did not change height within the accuracy of the measurements [3.3 ± 0.3 nm at 28 min (mean ± SD; $n = 47$), 3.5 ± 0.3 nm at 91 min ($n = 47$), 3.5 ± 0.3 nm at 208 min ($n = 47$), and 3.5 ± 0.3 nm at 253 min ($n = 50$)]. The fact that these heights of the GSDMD^Nterm oligomers are indistinguishable from each other and from the heights measured of arc-, slit-, and ring-shaped oligomers in various lipid mixtures (Appendix Tables S1 and S2) suggests that the dynamic pore-forming process of GSDMD^Nterm is not associated with a vertical height change of the GSDMD^Nterm oligomers. GSDMD^Nterm thus features a pore growing mechanism in the absence of a vertical collapse.

## Sub-nanometer topographs of GSDMD^Nterm oligomers

To further investigate the structural assembly of GSDMD^Nterm monomers, we recorded high-resolution AFM topographs of membrane-bound GSDMD^Nterm oligomers. We made SLMs from POPC, DOPG, DOPS, DOPE, and CL (40:20:10:20:10 molar ratio), which we incubated with GSDMD and caspase-1 for 60 min at 37°C. The AFM topographs showed subunits of the arc-, slit-, and ring-shaped GSDMD^Nterm oligomers (Fig 5A). On average, the subunits, which regularly distributed along the arc-, slit-, and ring-like oligomers (Fig 5A–C), showed lateral distances of 2.3 ± 0.3 nm (mean ± SD; $n = 117$) and protruded 3.5 ± 0.3 nm (mean ± SD; $n = 187$) from the lipid surface. On average, arc-like oligomers were assembled from 16.1 ± 0.4 (mean ± SE, $n = 26$) subunits, slit-like oligomers from 20.7 ± 0.4 ($n = 46$) subunits, and ring-like oligomers from 30.2 ± 0.2 ($n = 49$) subunits (Fig 5D–F). Whereas the arc- and slit-shaped oligomers formed pores having mean areas of 54.1 ± 2.8 nm$^2$ (mean ± SE, $n = 265$), the pores of ring-like oligomers showed mean areas of 158.1 ± 4.3 nm$^2$ ($n = 179$). In the absence of any structural data of the GSDMD^Nterm monomer, the subunits may be assigned to monomers forming the oligomer. Together with the observation that each of the oligomeric shapes can form transmembrane pores, it appears that the oligomeric GSDMD^Nterm subunits and perhaps even the monomers penetrate through the membrane.

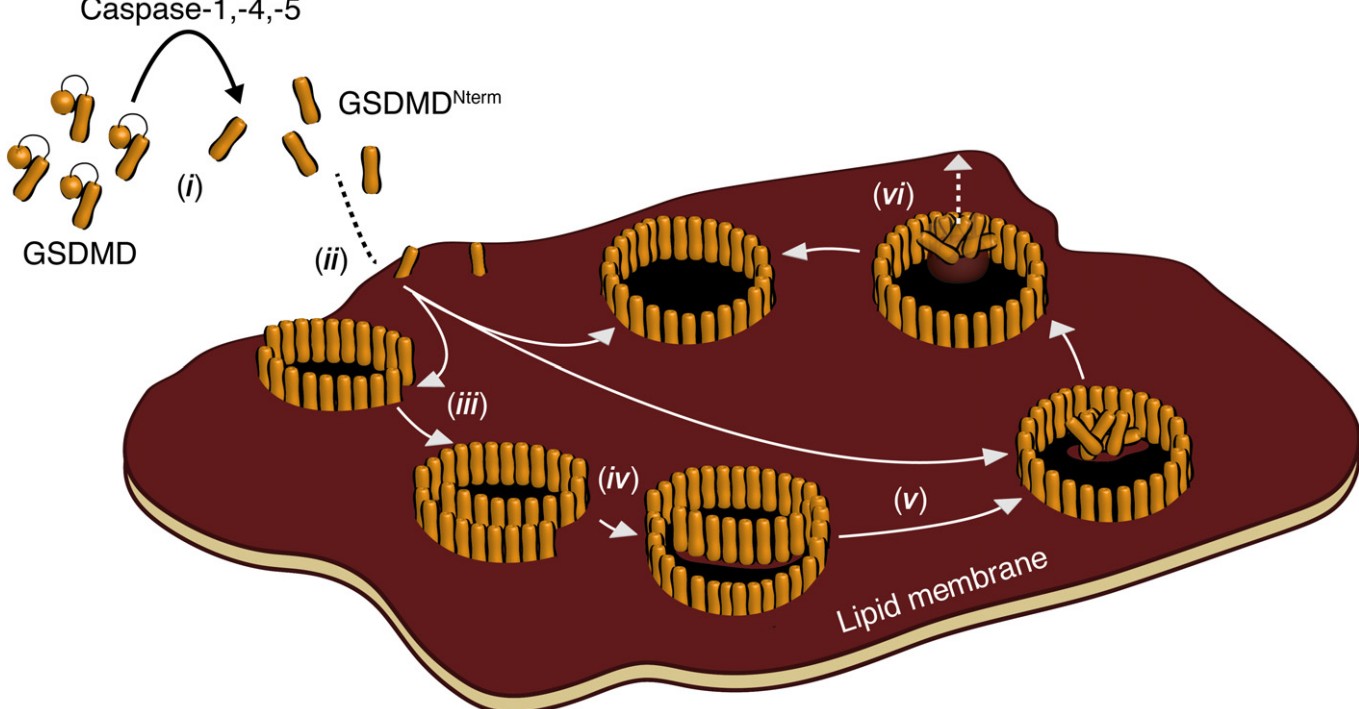

**Figure 6.  Model of GSDMD^Nterm oligomerization and pore formation.**
(i) Human GSDMD is cleaved by inflammatory caspase-1, caspase-4, or caspase-5. (ii) Afterwards, the cleaved N-terminal GSDMD domain GSDMD^Nterm binds to the lipid membrane where it (iii) oligomerizes into slit-, arc-, and ring-shaped structures. (iv) Arc- and slit-shaped oligomers can fuse into larger ring-shaped oligomers. Alternatively, oligomers may grow by assembling GSDMD^Nterm molecules to their free ends. During this process, GSDMD^Nterm oligomers do not significantly change height above the lipid membrane, suggesting the absence of large conformational changes. While each of the oligomeric species can form lytic transmembrane pores, the slit-, and arc-shaped structures fuse into ring-shaped structures in a time-dependent manner. This suggests that ring-shaped oligomers are thermodynamically more stable than the other oligomeric species. (v) Upon fusing into larger oligomers, GSDMD^Nterm and lipids may remain trapped inside the oligomer. (vi) With time, the trapped material loosens from the membrane, increases height, and exits to the solution.

## Discussion

Based on the structural insights revealed using high-resolution AFM, we derive a mechanistic model for the process of GSDMD[Nterm] assembling transmembrane pores (Fig 6). Recent publications on GSDMD[Nterm] oligomers have shown their ring and pore size to vary (Ding *et al*, 2016; Sborgi *et al*, 2016). However, we here observe that independently of whether human GSDMD has been cleaved by caspase-1, caspase-4, or caspase-5, the cleaved N-terminal domain binds and inserts into lipid membranes, where it assembles into a variety of arc-, slit-, and ring-shaped oligomers. Thereby, each of the oligomeric shapes can form membrane-penetrating, lytic pores. We further found that GSDMD[Nterm] binds to a wide range of lipid membrane compositions (Appendix Table S3), where it assembles arc-, slit-, and ring-shaped oligomers and pores. This observation shows that the assembly mechanism of GSDMD[Nterm] into oligomers having different shapes and sizes and the subsequent formation of pores is a robust and intrinsic property of GSDMD[Nterm]. However, whereas phosphoinositide PI(4,5)P2 increased the binding of GSDMD[Nterm] to the membrane, cholesterol reduced it. The effect of PI(4,5)P2 in facilitating GSDMD[Nterm] binding was expected, as phosphoinositide is present in the cytosolic leaflet of the plasma membrane (Di Paolo & De Camilli, 2006), which is the physiological target of GSDMD, and confirms earlier GSDMD[Nterm] binding experiments using liposomes and lipid strips (Ding *et al*, 2016; Liu *et al*, 2016). The effect of cholesterol in reducing GSDMD[Nterm] binding confirms previous binding experiments on cholesterol-containing membranes (Ding *et al*, 2016; Sborgi *et al*, 2016), but appears less obvious. Our results show that in the presence of PI(4,5)P2, concentrations of 15% and of 30% cholesterol progressively suppress GSDMD[Nterm] binding to the membrane. However, taken the complex lipid composition of the plasma membrane (van Meer *et al*, 2008; Liu *et al*, 2017), one may speculate that *in vivo* the lowering effect of cholesterol is compensated by the complex mixture of various lipids, by their distribution, and by the presence of membrane proteins. This hypothesis is supported by our results showing that GSDMD[Nterm] binds much more frequently to SLMs made from *E. coli* polar lipid extract supplemented with 30% cholesterol compared to SLMs made from POPS, POPC, DOPE, PI(4,5)P2, and supplemented with 30% cholesterol.

We found that the arc- and slit-shaped GSDMD[Nterm] oligomers represent dynamical structures that grow by the addition of further monomers and/or oligomers to their free ends. This growth changes the oligomeric shape. Once the open ends of arc- and slit-shaped oligomers formed a ring-shaped structure, most of the oligomers did not grow anymore. This observation leads to the assumption that ring-like GSDMD[Nterm] oligomers are the thermodynamically stable product of the assembly mechanism, while slit- and ring-like structures represent intermediate states. Occasionally, the pores of GSDMD[Nterm] ring-like oligomers contained some material, which presumably resulted from lipids and GSDMD[Nterm] molecules that could not be integrated during formation of the ring. Our time-lapse AFM topographs showed that this material could exit from the pore lumen, leaving an open transmembrane pore behind. Over time, many of the stable GSDMD[Nterm] oligomers formed open transmembrane pores.

The gasdermin protein family is conserved in vertebrates and represents an essential mediator of pyroptosis in human and murine cells. Similar to GSDMD, it has been described that also other GSDM family members, including GSDMA, GSDMA3, and GSDME can form transmembrane pores after cleavage (Ding *et al*, 2016; Rogers *et al*, 2017; Wang *et al*, 2017). Due to the structural conservation and functional similarity, it may thus be expected that the binding, insertion, and assembly mechanism of GSDMD[Nterm] described here and which is linked to pore formation are not only specific to GSDMD but describe a more general mechanism for other members of the gasdermin family.

## Materials and Methods

### Cloning, expression, and purification of GSDMD and caspases

Human GSDMD was cloned, expressed, and purified as described (Sborgi *et al*, 2016). Briefly, full-length human GSDMD with an N-terminal His$_6$-SUMO-tag was overexpressed in BL21(DE3) *E. coli* strains. Cells were harvested and disrupted by high-pressure microfluidization. SUMO-GSDMD was isolated by Ni$^{2+}$-NTA affinity purification, and the SUMO-tag was cleaved by ULP1 protease. The protein was further purified by ion-exchange and size-exclusion chromatography. Human caspase-1 was cloned, expressed in inclusion bodies in *E. coli*, and refolded as described (Ramage *et al*, 1995). Briefly, inclusion bodies were solubilized in 8 M urea, and the caspase was refolded by rapid dilution. During this refolding, the enzymes auto-activated. A gel filtration column was run to purify the final product and the concentration of active caspase determined by active site titration. Active recombinant human caspase-4 and caspase-5 were purchased from BioVision Inc, USA.

### Preparation of liposome and defect-free supported lipid membranes (SLMs)

Unilamellar liposomes were prepared by hydration of lipid films and extrusion through polycarbonate filters with 0.1 μm pore diameter (Nuclepore Polycarbonate, Whatman), according to the method described by Avanti Polar Lipids (www.avantilipids.com). The liposomes were prepared using *E. coli* polar lipid extract, *E. coli* polar lipid extract supplemented with cholesterol at weight ratio 70:30, POPG, POPC, or mixtures of POPC, DOPG, DOPS, DOPE, and CL (40:20:10:20:10 molar ratio); POPS, DOPE, and POPC (35:25:40 molar ratio); POPS, DOPE, and POPI (35:25:40 molar ratio); POPS, DOPE, and PI(4,5)P2 (35:25:40 molar ratio); POPS, POPC, DOPE, and PI(4,5)P2 (35:30:25:10 molar ratio); POPS, POPC, DOPE, PI(4,5)P2, and cholesterol (30:26:21:8:15 molar ratio); and POPS, POPC, DOPE, PI(4,5)P2, and cholesterol (24:21:18:7:30 molar ratio). Lipids and extruding equipment used for liposome preparation were purchased from Avanti Polar Lipids.

Supported lipid membranes were prepared by fusion of unilamellar liposomes on mica as previously described (Richter *et al*, 2006; Mulvihill *et al*, 2015). Briefly, the liposomes were sonicated (25 kHz, 500 W, Transsonic TI-H-5, ELMA) at room temperature in buffer solution (150 mM NaCl, 25 mM Hepes, pH 7.25) for 5 min and immediately adsorbed onto freshly cleaved mica at room temperature. After ≈ 60 min of adsorption, the SLMs were gently rinsed with buffer solution (50 mM NaCl, 100 mM Hepes, 5 mM TCEP, pH 7.4). SLMs were prepared freshly and imaged by AFM before being incubated with the proteins to ensure that SLMs were

defect-free and completely covered the AFM support (Appendix Fig S1). Only if the SLM covering the mica support showed no defects, such as holes, cracks, or stacked bilayers, GSDMD and caspases were applied. In order to verify our protocol for preparing defect-free SLMs, we dried some of the SLMs after AFM imaging. Drying for 1 min in air generated holes in the SLM and verified that the mica was covered by a SLM.

### Incubation of SLMs with GSDMD and caspases

After AFM imaging and verifying that the SLMs prepared were defect-free, they were incubated with 0.5 μM GSDMD and either 0.1 μM caspase-1, 2 units caspase-4, or 3 units caspase-5 at 37°C in buffer solution (50 mM NaCl, 100 mM Hepes, 5 mM TCEP, pH 7.4). After an incubation time of 60 min, the SMLs were washed and imaged in buffer solution (150 mM NaCl, 100 mM Hepes, pH 7.4) using AFM. For time-lapse AFM, SLMs were incubated with 0.7 μM GSDMD and 0.1 μM caspase-1 in buffer solution (150 mM NaCl, 100 mM Hepes, pH 7.4) at 37°C in the AFM fluid cell. AFM images of a selected area of the SLM were acquired every $\approx$ 9 min for more than 250 min. Buffer solutions were freshly made using nanopure water (18.2 M$\Omega$ cm$^{-1}$) and pro-analysis (> 98.5%) purity grade reagents from Sigma-Aldrich and Merck. Every experimental condition was reproduced at least three times using freshly prepared SLMs, new GSDM samples, and new AFM supports and cantilevers.

### Force–distance curve-based AFM (FD-based AFM)

FD-based AFM (Dufrene *et al*, 2013, 2017) was performed with a Nanoscope Multimode 8 (Bruker, USA) operated in PeakForce Tapping mode. The AFM was equipped with a 120-μm piezoelectric scanner and fluid cell. The two different AFM cantilevers used either had a nominal spring constant of 0.1 N m$^{-1}$, resonance frequency of $\approx$ 110 kHz in liquid, and a sharpened silicon tip with a nominal radius of 8–10 nm (BioLever mini BL-AC40, Olympus Corporation) or a spring constant of 0.4 N m$^{-1}$, a resonance frequency of $\approx$ 165 kHz in liquid, and a sharpened silicon tip with a nominal radius of $\approx$ 1 nm (PEAKFORCE-HiRs-F-A, Bruker Nano Inc., USA). BL-AC40 AFM cantilevers were calibrated using thermal tuning method (Hutter & Bechhoefer, 1993) and by ramping the cantilever on the mica support. To prevent the damage of the ultra-sharp tip having a nominal radius of $\approx$ 1 nm, we used the calibration the manufacturer provided for PEAKFORCE-HiRs-F-A cantilevers. AFM topographs were recorded in imaging buffer (150 mM NaCl, 100 mM Hepes, 5 mM TCEP, pH 7.4) as described (Pfreundschuh *et al*, 2014). Briefly, the maximum force applied to image the samples was limited to $\approx$ 100 pN, and the oscillation frequency and oscillation amplitude of the cantilever were set to 2 kHz and 40 nm, respectively. The AFM was placed inside a home-built temperature-controlled acoustic isolation box. FD-based AFM topographs were recorded at room temperature with exception of the time-lapse AFM experiments, which were recorded at 37°C. Topographs were analyzed using the AFM analysis software (NanoScope Analysis 1.5 r2).

### Transmission electron microscopy (TEM)

For transmission electron microscopy (TEM), unilamellar liposomes made from *E. coli* polar lipid extract were used at 1 mg ml$^{-1}$ concentration. Liposomes were incubated overnight at 37°C with 5 μM GSDMD and catalytic amounts of 1 μM caspase-1. 5 μl of the sample was pipetted onto a parlodion and carbon-coated glow-discharged copper grid and adsorbed for 1 min. The grid was then washed with four droplets of nanopure water and subsequently negatively stained with 2% uranyl acetate for 10 s, blotting between each step. Grids were imaged using a TEM (FEI T12 equipped with a 120-kV LaB6 filament, FEI Company, the Netherlands) operated at 120 kV. Images were recorded using a bottom-mounted CMOS camera (TemCam-F416, TVIPS, Germany).

**Expanded View** for this article is available online.

### Acknowledgements

We thank the BioEM-Lab of the Biozentrum of the University of Basel for assistance with TEM imaging and J. Pipercevic for technical assistance. This work was funded by the Swiss National Science Foundation (Grant 205320_160199) and the European Union Marie Curie Actions Program through the ACRITAS Initial Training Network (FP7-PEOPLE-2012-ITN, Project 317348).

### Author contributions

Experiments were designed by EM and DJM. GSDMD samples and some of the caspase were provided by LS and SH. Sample preparation for AFM experiments and the AFM experiments were conducted by EM and SAM. AFM experiments were analyzed by EM and by MP. TEM sample preparation and imaging were done by SAM. EM, MP, and DJM realized the conclusive figure. All authors wrote the manuscript.

### Conflict of interest

The authors declare that they have no conflict of interest.

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
