## [Review Process File · The EMBO Journal]

Mechanism of membrane pore formation by human Gasdermin-D

Estefania Mulvihill, Lorenzo Sborgi, Stefania A. Mari, Moritz Pfreundschuh, Sebastian Hiller & Daniel J. Müller.

Review timeline:

Submission date:	30 th September 2017
Editorial Decision:	2 nd November 2017
Revision received:	1 st April 2018
Editorial Decision:	30 th April 2018
Revision received:	2 nd May 2018
Accepted:	7 th May 2018

Editor: Karin Dumstrei.

Transaction Report:

1st Editorial Decision

2nd November 2017

Thank you for submitting your manuscript to The EMBO Journal. Your study has now been seen by three referees and their comments are provided below.

As you can see from the comments, the referees appreciate the AFM imaging of pore formation and the insights provided. However, they also find that the analysis has to be extended in order to consider publication here. They offer a number of constructive comments for how to do so.

Should you be able to address the concerns raised then I would like to invite you to submit a revised version. I should add that it is EMBO Journal policy to allow only a single round of revision and that it is important to resolve the major concerns at this stage

REFeree REPORTS.

Referee #1:

A milestone progress in innate immune research in the past few years is the identification of Gasdermin-D (GSDMD) as the pyroptotic substrate of caspase-1 and caspase-11 in response various infectious signals. GSDMD executes pyroptosis via an intrinsic pore-forming activity in its N-terminal domain, which clarifies the nature of pyroptotic cell death. GSDMD belongs to a large Gasdermin family that appears to share the pore-forming domain despite that their physiological function and activation mechanism remain largely unknown. Following these findings, it is of great interest in the innate immune field to understand the mechanism how GSDMD forms pores and disrupts the membrane. Research on this topic is also grabbing attentions from the cell death field as well as the big community studying the membrane pore formation processes. In the submitted manuscript, Mulvihill et al. dissect the pore-forming process of GSDMD on the membrane by AFM and provide valuable insights into the underlying mechanism of GSDMD pore formation. However,

there are still some concerns that need to be addressed before the manuscript can be accepted for publication.

1) GSDMD-N domain has been shown to specifically bind to phosphoinositides and cardiolipin as well as phosphatidylserine. Phosphoinositides are known to be present in mammalian cell membrane, particularly the cytoplasmic leaflet of plasma membrane. Upon binding to phosphoinositides, GSDMD-N domain forms extensive pores on the membrane, disrupting the membrane integrity and eventually triggering cell pyroptosis. Compared with cardiolipin, phosphoinositides are the more physiological relevant targets of GSDMD as well as other Gasdermin-family members. Therefore, the authors should also exhibit the pore formation on membranes containing phosphoinositides. The proposed negative effect of cholesterol could also be tested on phosphoinositides-containing membranes. Additionally, phosphatidylserine-containing membrane can be assayed and the results will be of great interest to the field. These control assays will make the study more complete and the results obtained are more physiologically relevant.

2) In the time-lapse AFM analysis that tracks the process of GSDMD-N domain oligomerization and pore formation, the authors showed that the heights of oligomeric GSDMD-N domain protruding from the membrane surface did not change during pore formation. This observation indicates a "growing" process instead of involving a "prepore-to-pore transition" step seen with many other pore-forming proteins. This observation is interesting but needs to be strengthened by additional evidences. For example, the author could check whether a possible prepore of GSDMD-N domain could be induced in the solution state by adding phosphoinositide into caspase-cleaved GSDMD. Also a typical cholesterol-dependent cytolysin, which is known to form prepores in solution and undergo a vertical collapse on the membrane, could be included as a control.

3) Since other Gasdermin-family members also harbor the pore-forming activity, it is important to know whether a similar dynamic process of Gasdermin N-domain assembling into transmembrane pores also applies to other gasdermins. This data will increase the impact of the current study.

Referee #2:

Mulvihill and al present an AFM study to characterize the mechanism of pore formation by inflammatory Gasdermin-D in model membrane systems. They study the role of lipid composition on GSDMD binding to the membrane and pore formation and provide a time sequence for GSDMD assembly into rings. This work presents some new details about how lipids influence GSDMD binding and assembly in the membrane. Based on their data they conclude that cholesterol plays an inhibitory role in helping GSDMD binding to the membrane. Moreover, they provide high imaging resolution of GSDMD pores and some evidence about the kinetics of GSDMD pore formation, where GSDMD seems to pass through slits/arcs intermediates before forming rings that, according to the authors, represent a more thermodynamically stable assembling of oligomerized GSDMD. However, in my opinion this work represents an incremental advance on the previous work by the same authors in Sborgi et al *Embo J.* 2016, where many of the results and ideas presented in this AFM study have been previously proposed. That said, I would like to point out that the quality of the AFM imaging is impressive and offers the potential for a more in depth analysis of GSDMD pore formation beyond what is presented here and that would make a difference in terms of novelty of the findings.

For example, the idea of a kinetic mechanism for GSDMD pore formation is intriguing; however the conclusions should be supported by more robust evidence. The authors should provide statistical numbers of how many slit and arcs shapes turn into rings, how many rings form pores over time, whether the pore size increases over time before reaching the final size, whether slit and arcs pores are smaller than ring pores and whether increasing concentration of GSDMD affect all these parameters. They could also analyze what is the smallest size of slit and arc that can form a pore, and more interestingly, given the high quality of the resolution imaging, what is the minimum number of GSDMD molecules that can be associated with a membrane pore.

Based on the relatively wide distribution of the ring diameter, the authors claim that GSDMD pore formation is a "structurally rather flexible process" (pag 6). This can be a result of a dynamic assembling process where the pore grows over time before reaching a steady state. It would be interesting to explore this hypothesis by, again, looking at the pore size over time, also in the context of understanding the nature of GSDMD pores (only protein or lipid-protein based). Still, for a better

understanding of the kinetics mechanism, it would be useful to plot in a graph how the number of slit/arcs and rings change over time. According to what the authors observe (pag 8) slit/arcs should reach a plateau while rings should constantly increase.

Another important aspect is the first part of the manuscript regarding the analysis of lipid composition. The authors should specify why the choice of the POPC, DOPG, DOPS, DOPE, and CL (40:20:10:20:10 w/w) lipid composition in their experiments. GSDMD has been shown to form pores in the inner leaflet of the plasma membrane that are responsible for Pyroptosis execution. The authors should analyze GSDMD pore activity in a lipid composition mimicking this leaflet, which by the way, contains about 30% cholesterol. In this sense, the authors should explain how come cholesterol inhibits pore formation and how can this be put into perspective given the presence of cholesterol in the physiological target membrane of this protein.

The paragraph describing the sub-nanometer topographs of GSDMD is shallow. The authors refer to substructures, but it's not clear to me what they are referring to. They should illustrate with arrows in the figures. What do they mean with lateral distance? Again an illustration would be useful.

Minor aspects:

- For clarity, the authors should provide a zoom in picture of a slit, an arc and a ring
- The title of the first paragraph in the results section is misleading because the authors said that GSDMD oligomerization depends on the lipid composition, however later on in the text they specify that GSDMD assembly is not sensitive to the lipid composition (pag 6). They should write again the title focusing more on the Cholesterol role rather than lipids in general.
- On page 5 the authors, based on the low height of the structures protruding the membrane, conclude that GSDMD monomers fully insert into the membrane. This is a quite strong statement not supported by high resolution structural data
- Fig 2A provides the same information of fig 1A and graph G in fig 2 is the same than graph G in fig 1
- Figure s3 does not provide any additional information (see fig 3)
- In the author contribution section some authors are not mentioned

Referee #3:

Mulvihill study the insertion and pore-formation of GSDMD N-termini into membranes by AFM. They find that the structures formed are of different shapes and sizes. The AFM pictures are impressive and of very high quality and resolution. Taken that this is the strong part of the manuscript, the most should be made of this, e.g., the scale of height should be added and interpreted in all pictures. The main criticism is that it is difficult to judge how relevant the data is since a membrane on mica might not allow all conformations the protein might assume in a real membrane.

Some important points are listed below.

The dimensions of the pore should be quantified not only by diameter but also by the number of GSDMD subunits per pore, which seem possible, at least from the data in figure 4 and is the strong point of this work.

More detailed statistics on the shape and diameter of the pores should be presented. Is there a most-common diameter / number of subunits of the ring-shaped pore that is the most stable, or do the pores just keep growing?

We would have expected more orderly structures. Using this experimental system, do other, better studied pore-forming proteins also assume such divers pore shapes? How representative is this of the physiological structure?

Why do they only test the E. coli membrane with and without cholesterol, not also the synthetic membrane?

What happens if GSDMD and Casp1 without the membrane being around and later added to the membrane? Is it important that the GSDMD N-term is "fresh"?

Figure 2: do the different caspases cleave GSDMD at different sites? If not, the result is highly expected. The authors provide no indication on why this was an interesting question for them to test.

The authors claim that "the relatively wide distribution of the diameter of ring-shaped oligomers and the coexistence of arc-, slit-, and ring-shaped oligomers indicates that the assembly of transmembrane pores by GSDMDNterm is a structurally rather flexible process." - How can the exclude that these phenomena are not a result of their experimental settings and that the natural pore form in a more orderly fashion?

The protrusion (Z-Axis) should be depicted separately and quantified not only in Figure 1.

The use of supported lipid membranes on mica (a solid carrier) could be a problem if the GSDMD pore involves GSDMD molecules protruding from the membrane on the outer side. The authors should comment how this is dealt with in the field for other pore-forming proteins

In Figure 4, the central part in some pores is obviously protruding higher than the surrounding ring. Why is this not quantified? Could this be a consequence of a secondary structure trying to form (that might be able to stabilize the ring and stop its growth)? The presence of GSDMD in the middle of the pore is even depicted in Figure 5 (final stage), but not sufficiently discussed. However, the model shown does not properly represent the obtained data because the different height of the protein in the center of the pore in the terminal stage is not shown.

If the authors want to reach the broad readership of the EMBO Journal, they should explain why they think their synthetic membrane is physiologically relevant and how it is composed. The abbreviations are not explained anywhere.

Why do the authors keep repeating the composition of their membrane including the rations throughout the manuscript if it is always the same? It is enough the just say "SLM" after it was first mentioned, unless it differs.

In the first sentence, "been" is missing, or it should be "was" instead of "has".

EMBOJ-2017-98321 "Mechanism of membrane pore formation by human Gasdermin-D"

Point-by-point response to the comments of reviewer #1

Reviewer #1: *A milestone progress in innate immune research in the past few years is the identification of Gasdermin-D (GSDMD) as the pyroptotic substrate of caspase-1 and caspase-11 in response various infectious signals. GSDMD executes pyroptosis via an intrinsic pore-forming activity in its N-terminal domain, which clarifies the nature of pyroptotic cell death. GSDMD belongs to a large Gasdermin family that appears to share the pore-forming domain despite that their physiological function and activation mechanism remain largely unknown. Following these findings, it is of great interest in the innate immune field to understand the mechanism how GSDMD forms pores and disrupts the membrane. Research on this topic is also grabbing attentions from the cell death field as well as the big community studying the membrane pore formation processes. In the submitted manuscript, Mulvihill et al. dissect the pore-forming process of GSDMD on the membrane by AFM and provide valuable insights into the underlying mechanism of GSDMD pore formation. However, there are still some concerns that need to be addressed before the manuscript can be accepted for publication.*

Authors: Thank you for your encouraging and constructive comments, which guided us to revise our manuscript. Below please find in our point-by-point response how we addressed each of the specific concerns by conducting additional experiments and revising our manuscript.

Reviewer #1: *1) GSDMD-N domain has been shown to specifically bind to phosphoinositides and cardiolipin as well as phosphatidylserine. Phosphoinositides are known to be present in mammalian cell membrane, particularly the cytoplasmic leaflet of plasma membrane. Upon binding to phosphoinositides, GSDMD-N domain forms extensive pores on the membrane, disrupting the membrane integrity and eventually triggering cell pyroptosis. Compared with cardiolipin, phosphoinositides are the more physiological relevant targets of GSDMD as well as other Gasdermin-family members. Therefore, the authors should also exhibit the pore formation on membranes containing phosphoinositides. The proposed negative effect of cholesterol could also be tested on phosphoinositides-containing membranes. Additionally, phosphatidylserine-containing membrane can be assayed and the results will be of great interest to the field. These control assays will make the study more complete and the results obtained are more physiologically relevant.*

Authors: Since phosphoinositides are more physiological targets of GSDMD than cardiolipin, the reviewer suggests to characterize the formation of GSDMD on membranes containing phosphoinositides. He/she further suggests to test the proposed negative effect of cholesterol using phosphoinositide-containing

membranes. We followed these advices and conducted control experiments to test the effect of different phosphoinositide concentrations on the binding and assembly of GSDMD^{Nterm} (Fig. R1.1 and R1.2). Thereto, we first assembled SLMs from POPS, DOPE and POPC (35:25:40 molar ratio), from POPS, DOPE and POPI (35:25:40 molar ratio) and from POPS, DOPE and PI(4,5)P₂ (35:25:40 molar ratio) (Fig. R1.1). As described for other experiments (revised Fig. 1), each of the SLMs was incubated with 0.5 μM GSDMD and 0.1 μM caspase-1 for 60 min at 37°C. The AFM topographs showed that GSDMD^{Nterm} also bound to SLMs made from POPS, DOPE and POPC, where it assembled arc- slit- and ring-like oligomers (Fig. R1.1A,D,F). Again, the oligomers could form transmembrane pores. However, we could not observe GSDMD^{Nterm} binding to SLMs made from POPS, DOPE and POPI (Fig. R1.1B). Lastly, GSDMD^{Nterm} bound much more frequently to SLMs made from POPS, DOPE and PI(4,5)P₂ compared to SLMs made from POPS, DOPE and POPC (Fig. R1.1C). This finding, which indicates that PI(4,5)P₂ promotes the binding of GSDMD^{Nterm} to lipid membranes, is in agreement to recently published data characterizing GSDMD binding to liposomes (Ding *et al*, 2016). Our data also showed that the presence of PI(4,5)P₂ had no influence on the diameter of ring-like oligomers (Fig. R1.1I,J). However, since in PI(4,5)P₂-containing SLMs the occurrence of arc- and slit-like oligomers reduced and of ring-like oligomers increased, one may conclude that PI(4,5)P₂ promotes the assembly of ring-like oligomers. The new data has now been included into our revised manuscript and discussed (see new Fig. 2, new Results, section ‘Opposing roles of phosphoinositide and cholesterol’, and revised Discussion).

Figure R1.1, included as new Fig. 2. Effect of phosphoinositide on the assembly of GSDMD^{Nterm} oligomers and pores. (A-C) AFM topographs showing GSDMD^{Nterm} oligomers formed on SLMs made from (A) POPS, DOPE and POPC (35:25:40 molar ratio), (B) POPS, DOPE and POPI (35:25:40 molar ratio), and (C) POPS, DOPE and PI(4,5)P₂ (35:25:40 molar ratio). (D-F) High-magnification AFM topographs of ring-like oligomers with corresponding height profiles. (G-H) High-magnification AFM topographs of arc- and slit-like oligomers with corresponding height profiles. (I, J) Histograms showing the relative count of ring-like oligomers for the two lipid compositions, with mean diameters of 25.8 ± 1.6 nm (n < 20) and 26.7 ± 0.5 nm (n < 112). (K) Bar chart showing the relative count of arcs, slits, and rings for the two lipid compositions.

ratio), and (C) POPS, DOPE and PI(4,5)P₂ (35:25:40 molar ratio). (D and E) Topographs showing arc-, slit- and ring-like GSDMD^{Nterm} oligomers formed on SLMs assembled from (D) POPS, DOPE and POPC (35:25:40 molar ratio) and (E) POPS, DOPE and PI(4,5)P₂ (35:25:40 molar ratio). (F-H) Height profiles of GSDMD^{Nterm} oligomers measured along the red lines indicated in the topographs (A-C). Black dashed lines indicate the SLM surface (0 nm height). (I-J) Diameter of GSDMD^{Nterm} ring-shaped oligomers formed on SLMs made from (I) POPS, DOPE and POPC and (J) POPS, DOPE and PI(4,5)P₂. Thin black lines are Gaussian fits determining the mean \pm SE values of each distribution. (K) Number of arc-, slit- and ring-shaped GSDMD^{Nterm} oligomers formed on SLMs made from POPS, DOPE and POPC (black bars) or POPS, DOPE and PI(4,5)P₂ (grey bars). Bars represent averages and error bars SD. The full color range of the topographs corresponds to a vertical scale of 12 nm. Scale bars, 100 nm (A-C), 20 nm (D and E) and 10 nm (F-H). Averages and errors are given in the text and summarized in **Appendix Table S1**.

The reviewer further suggested to investigate the effect of cholesterol in phosphoinositide-containing membranes on GSDMD^{Nterm} binding and assembly. We conducted experiments to characterize the effect of cholesterol in the presence of PI(4,5)P₂ (**Fig. R1.2**). For our experiments we assembled SLMs from POPS, POPC, DOPE and PI(4,5)P₂ (35:30:25:10 molar ratio) and from the same lipid mixture supplemented with either 15% or 30% of cholesterol (molar ratio). Each of the SLMs was incubated with 0.5 μ M GSDMD and 0.1 μ M caspase-1 for 60 min at 37°C. Compared to SLMs having no cholesterol, AFM topographs showed less GSDMD^{Nterm} oligomers on SLMs containing 15% cholesterol (**Fig. R1.2A-D**). Finally, at cholesterol concentrations of 30% the binding of GSDMD^{Nterm} was mostly suppressed (**Fig. R1.2E,F**). The new data has been included into our revised manuscript and discussed (see **new Appendix Fig. S6, revised Results**, section 'Opposing roles of phosphoinositide and cholesterol', and **revised Discussion**).

Figure R1.2, included as new **Appendix Fig. S6**. Effect of cholesterol on the assembly of GSDMD^{Nterm} oligomers in phosphoinositide-containing lipid membranes. Overview and high-

resolution AFM topographs showing GSDMD^{Nterm} oligomers formed on SLMs made from (A and B) POPS, POPC, DOPE and PI(4,5)P2 (35:30:25:10 molar ratio), (C and D) POPS, POPC, DOPE, PI(4,5)P2 and cholesterol (30:26:21:8:15 molar ratio), and (E and F) POPS, POPC, DOPE, PI(4,5)P2 and cholesterol (24:21:18:7:30 molar ratio). The full color range of the topographs corresponds to a vertical scale of 5 nm. Scale bars, 100 nm (A, C and E) and 50 nm (B, D, and F).

Furthermore, the reviewer suggested to test the effect of phosphatidylserine-containing membranes on GSDMD^{Nterm} binding and assembly. We kindly refer to **Figs. 1 and 2** and **Appendix Fig. S6** of our manuscript, in which we characterized GSDMD^{Nterm} binding and assembly on phosphatidylserine-containing (DOPS and POPS) membranes. We have carefully revised our manuscript to explain the experiments more clearly (see **revised Results**).

Reviewer #1: 2) *In the time-lapse AFM analysis that tracks the process of GSDMD-N domain oligomerization and pore formation, the authors showed that the heights of oligomeric GSDMD-N domain protruding from the membrane surface did not change during pore formation. This observation indicates a "growing" process instead of involving a "prepore-to-pore transition" step seen with many other pore-forming proteins. This observation is interesting but needs to be strengthened by additional evidences. For example, the author could check whether a possible prepore of GSDMD-N domain could be induced in the solution state by adding phosphoinositide into caspase-cleaved GSDMD. Also a typical cholesterol-dependent cytolysin, which is known to form prepores in solution and undergo a vertical collapse on the membrane, could be included as a control.*

Authors: The reviewer suggests to further strengthen our finding that GSDMD^{Nterm} forms pores in a "growing" rather than in a "prepore-to-pore transition" step. We made the following controls, as suggested by the reviewer, to test and strengthen our finding:

1) The reviewer suggests to test whether a possible prepore of GSDMD^{Nterm} could be induced in the solution state by adding phosphoinositide into caspase-cleaved GSDMD (GSDMD^{Nterm}). Unfortunately, phosphoinositide is not water-soluble and we couldn't find any commercially available lysophosphatidylinositide (which could possibly be slightly water-soluble). To test whether GSDMD^{Nterm} can form prepores before inserting into the lipid membrane we incubated GSDMD with caspase in the absence of any lipids at 37°C for 60 min, which was the incubation time used in our AFM-assays, and for much extended time overnight. After this incubation, we imaged the sample by AFM (**Fig. R1.3B**, included as **new Appendix Fig. S3**) and by TEM (**Fig. R1.4C**, included as **new Appendix Fig. S4**). None of the controls showed that cleaved GSDMD (GSDMD^{Nterm}) pre-formed arc-, slit- or ring-shaped oligomers in the absence of lipid membranes. The new data have been included into our revised manuscript (see revised Manuscript, **new Results** section 'GSDMD^{Nterm} oligomerization in liposomes', and **new Appendix Figs. S3 and S4**).

2) We further incubated liposomes with GSDMD in the absence (**Fig. R1.4**) and in the presence of caspase-1 (**Fig. R1.5**, included as **new Appendix Fig. S5**). Only in

the presence of both GSDMD and caspase-1, we observed GSDMD^{Nterm} inserting into liposomes and forming arc-, slit- and ring-like oligomers. The TEM images showing arc-, slit- or ring-shaped GSDMD^{Nterm} oligomers in liposomes indicate that this effect is not restricted to SLMs. The new data has been included into our revised manuscript (see **new Results** section 'GSDMD^{Nterm} oligomerization in liposomes' and **new Appendix Figs. S5**).

Figure R1.3, included as new Appendix Fig. S3. AFM of human GSDMD incubated with caspase-1 in the absence of a lipid membrane on mica or incubated in the absence of caspase-1 on a SLM. (A) AFM topograph of freshly cleaved atomically flat mica support imaged in buffer solution (50 mM NaCl, 100 mM Hepes, 5 mM TCEP, pH 7.4). (B) Topograph of GSDMD (0.5 μ M) and caspase-1 (0.1 μ M) incubated in buffer solution (50 mM NaCl, 100 mM Hepes, 5 mM TCEP, pH 7.4) for 60 min at 37°C on the mica support. (C) Topograph of a SLM made from *E. coli* polar lipid extract and incubated with GSDMD (0.5 μ M) in the absence of caspase in buffer solution (50 mM NaCl, 100 mM Hepes, 5 mM TCEP, pH 7.4) for 60 min at 37°C. The full color range of the topographs corresponds to a vertical scale of 4.5 nm. Scale bars, 100 nm.

Figure R1.4, included as new Appendix Fig. S4. TEM of liposomes, of liposomes incubated with human GSDMD in the absence of caspase-1, and of human GSDMD incubated with caspase-1. (A) TEM image of liposomes made from *E. coli* polar lipid extract. During the adsorption onto the TEM grid liposomes fused with each other forming lipid membranes. Liposomes made from other lipids used in this work appeared very similar (not shown here). (B) TEM image of liposomes made from *E. coli* polar lipid extract and incubated with 5 μ M GSDMD in the absence of caspase-1. None of the lipid membranes imaged ($n > 50$) showed arc-, slit- or ring-like structures. (C) TEM image of GSDMD (5 μ M) incubated with caspase-1 (0.1 μ M) overnight at 37°C in the absence of liposomes. The image shows no arc-, slit- or ring-like structures. Scale bars, 500 nm (A) and 100 nm (B and C).

Figure R1.5, included as new Appendix Fig. S5. TEM of arc-, slit- and ring-shaped GSDMD^{Nterm} oligomers on liposomes. (A and B) TEM images of liposomes incubated with GSDMD in the presence of caspase-1. Upon adsorption to the TEM grid most of the liposomes fused with each other forming lipid membranes. GSDMD^{Nterm} oligomers inserted into lipid membranes are observed. (C and D) Higher resolution images of arc-, slit- and ring-shaped oligomers formed by GSDMD^{Nterm}. Arc-like oligomers are pointed out by single arrowheads and slit-like oligomers by double arrowheads. Suspended liposomes made from *E. coli* polar lipid extract were incubated over night at 37°C with 5 μ M GSDMD and 1 μ M caspase-1. Samples were negatively stained with uranyl acetate and imaged at 120 kV (**Materials and Methods**). Scale bars, 200 nm (A), 100 nm (B) and 50 nm (C and D).

The reviewer further suggests a control showing “a typical cholesterol-dependent cytolysin, which is known to form prepores and undergo a vertical collapse on the membrane”. In a previous paper, we have shown that our AFM assay, which we used here to characterize the pore formation of GSDMD, is sufficiently sensitive to follow the "prepore-to-pore transition" step of the typical cholesterol-dependent cytolysin pneumolysin (PLY) from *Streptococcus pneumoniae* (van Pee *et al*, 2016). Below we reproduce AFM images taken from this paper (**Fig. R1.6**), which show ring-shaped PLY oligomers binding to a lipid membrane and vertically collapsing from the higher protruding prepore state to the lower protruding pore state. In summary, these exemplified AFM images recording the vertical collapse of PLY on the membrane show that the identical AFM assay used in this work is sensitive to detect such vertical collapse of pore forming proteins, which was not observed for GSDMD.

Figure R1.6. Direct observation of pneumolysin (PLY) oligomerization, ring formation, vertical collapse and transmembrane pore formation. The time-lapse AFM topographs show the binding of PLY oligomers, ring formation, vertical collapse of the prepores and formation of transmembrane pores on SLMs (POPC and cholesterol with a molar ratio of 1:1). Times

indicate the minutes after addition of PLY to the imaging buffer. Height profiles of PLY oligomers (red curve lines) are measured along the red lines in the topographs at the indicated time points. PLY prepores are seen as higher protruding (brighter) rings in the topographs recorded at 282 and 342 min. Dashed black lines (0 nm) indicate the SLM surface and continuous black lines (≈ -5 nm) the surface of the mica support. The full color range of the AFM topographs recorded at 37°C correspond to a vertical scale of 25 nm. Scale bars of topographs correspond to 50 nm and of height profiles to 10 nm. The Figure has been extracted from Figure 4 from reference (van Pee *et al*, 2016).

Reviewer #1: 3) *Since other Gasdermin-family members also harbor the pore-forming activity, it is important to know whether a similar dynamic process of Gasdermin N-domain assembling into transmembrane pores also applies to other gasdermins. This data will increase the impact of the current study.*

Authors: To further increase the impact of the study the reviewer suggests to characterize whether other members of the Gasdermin family show a similar process of pore formation as observed for GSDMD. We have started to record some data of the Gasdermin family member Gasdermin-A3 (**Fig. R1.7**). The data shows that Gasdermin-A3 similarly to GSDMD assembles slit-, arc- and ring-like oligomers. No vertical differences of the Gasdermin-A3 oligomers could be observed, thus indicating the absence of a vertical collapse. The data also shows that the diameter of ring-like Gasdermin-A3 oligomers is much more constrained compared to the ring-like GSDMD oligomers. However, as our manuscript focuses on the oligomeric assembly and pore formation mechanism of human GSDMD, we would like to keep this focus throughout the paper. Extending the submitted work to other Gasdermin family members, such as Gasdermin-A3, would be beyond the scope of our study. After having discussed this query with the editor handling our paper at EMBO J, we have thus decided not to include this additional data showing the assembly and pore-forming mechanisms of Gasdermin-A3 into our paper. Once our paper has been accepted for publication, we will address this issue raised by the reviewer and search for similarities and individualities of the oligomeric assembly and pore forming mechanisms of other Gasdermin family members (including Gasdermin-A3).

Figure R1.7. Characterization of the assembly of GSDMA3^{Nterm}. AFM topograph showing GSDMA3^{Nterm} oligomers formed on a supported lipid membrane (SLM) assembled from *E. coli* polar lipid extract. The SLM was incubated with buffer solution (50 mM NaCl, 0.5 mM TCEP, 20 mM Hepes, pH 7.4) containing 38 μ M GSDMA3 and 7 μ M TEV protease for 120 min at 37 °C. After the incubation time, the SLM was rinsed and imaged in buffer solution (50 mM NaCl, 0.5 mM TCEP, 20 mM Hepes, pH 7.4). Green arrows indicate arc-like, white arrows slit-like and pink arrows ring-shaped oligomers. The full color range of the topograph corresponds to a vertical scale of 8.6 nm. Scale bar, 100 nm.

EMBOJ-2017-98321 “Mechanism of membrane pore formation by human Gasdermin-D”

Point-by-point response to the comments of reviewer #2

Reviewer #2: *Mulvihill and al present an AFM study to characterize the mechanism of pore formation by inflammatory Gasdermin-D in model membrane systems. They study the role of lipid composition on GSDMD binding to the membrane and pore formation and provide a time sequence for GSDMD assembly into rings. This work presents some new details about how lipids influence GSDMD binding and assembly in the membrane. Based on their data they conclude that cholesterol plays an inhibitory role in helping GSDMD binding to the membrane. Moreover, they provide high imaging resolution of GSDMD pores and some evidence about the kinetics of GSDMD pore formation, where GSDMD seems to pass through slits/arcs intermediates before forming rings that, according to the authors, represent a more thermodynamically stable assembling of oligomerized GSDMD. However, in my opinion this work represents an incremental advance on the previous work by the same authors in Sborgi et al Embo J. 2016, where many of the results and ideas presented in this AFM study have been previously proposed. That said, I would like to point out that the quality of the AFM imaging is impressive and offers the potential for a more in depth analysis of GSDMD pore formation beyond what is presented here and that would make a difference in terms of novelty of the findings.*

Authors: Thank you for your encouraging and critical comments, which guided us to revise our manuscript. The reviewer suggested to extend the analysis and to conduct a considerable amount of additional experiments, which the reviewer specified below. These experiments, together with the experiments suggested by the other reviewers, considerably advanced our study compared to previous works, strengthened the manuscript and our scientific findings. For a detailed explanation of how we addressed each of the specific concerns of the reviewer, we kindly refer to our point-by-point response.

Reviewer #2: *For example, the idea of a kinetic mechanism for GSDMD pore formation is intriguing; however the conclusions should be supported by more robust evidence. The authors should provide statistical numbers of how many slit and arcs shapes turn into rings, how many rings form pores over time, whether the pore size increases over time before reaching the final size, whether slit and arcs pores are smaller than ring pores and whether increasing concentration of GSDMD affect all these parameters. They could also analyze what is the smallest size of slit and arc that can form a pore, and more interestingly, given the high quality of the*

resolution imaging, what is the minimum number of GSDMD molecules that can be associated with a membrane pore.

Authors: The reviewer suggested to support the conclusions of our paper by more robust evidence. He/she suggested to provide statistical numbers of how many slits and arcs turn into rings. In our time-lapse AFM experiments we imaged supported lipid membranes (SLMs) while incubating them with a limited amount of GSDMD and caspase-1 at 37°C (see **Results**, section ‘Imaging GSDMD^{Nterm} oligomerization and pore formation’). Thus, during the experiments new GSDMD^{Nterm} could insert and assemble arc-, slit- and ring-like oligomers throughout recording time-lapse AFM images. We analyzed our time-lapse images and plotted the number of slits/arcs and rings over time (**Fig. R2.1, included as new Appendix Fig. S10**). The analysis shows that the number of arcs remains at low levels, the number of slits reduces with the time approaching lower levels, and the number of rings increases with time approaching a plateau. This finding indicates that arc- and slit-like oligomers, which continuously inserted from the buffer solution into the membrane, fused into more stable and larger ring-like GSDMD^{Nterm} oligomers. Furthermore, we have analyzed the time-lapse AFM topographs to count how many rings were formed from arc- and slit-shaped structures. We found, that $4.7 \pm 2.5\%$ (ave \pm SE, $n_{\text{Experiments}} = 3$, $n_{\text{Oligomers}} = 2718$) of the ring-like oligomers assembled from arc-like oligomers, $79.5 \pm 9.8\%$ ring-like oligomers assembled from slit-like oligomers and $24.8 \pm 7.4\%$ of the GSDMD^{Nterm} oligomers were already observed as rings (**Fig. R2.1**). It may be assumed, that the already assembled GSDMD^{Nterm} rings formed faster than the time resolution of the time-lapse topographs ($\approx 5\text{--}12$ min, see **Fig. 4, Appendix Figs. S7, and new Appendix Fig. S8**). Throughout the time-lapse experiments the number of ring-shaped oligomers GSDMD^{Nterm} increased while the number of arc-like oligomers remained at low levels and the number of slit-like oligomers decreased. We have included this information into our revised manuscript (see **revised Results**, section ‘Imaging GSDMD^{Nterm} oligomerization and pore formation’, **revised Discussion** and **new Appendix Figs. S7, S8 and S10**).

Figure R2.1, included as new Appendix Fig. S10. Relative count of arc-, slit- and ring-shaped oligomers over time. The numbers of arc-, slit- and ring- shaped oligomers were determined from time-lapse AFM topographs recording the GSDMD^{Nterm} oligomerization and pore formation (**Fig. 4, Appendix Fig. S7, and new Appendix Fig. S8**). The number of arc-, slit- and ring-like oligomers was normalized to 1. The total number of GSDMD oligomers analyzed was 1398 in (A), 1142 in (B) and 178 in (C).

The reviewer further asked whether the pore size increases over time before reaching the final size. We have analyzed our time-lapse AFM images to answer this question (**Fig. R2.2, included as new Appendix Fig. S9**). The data show that the pore size (area) formed by GSDMD^{Nterm} oligomers increases with time and approaches a plateau. At this point, most the GSDMD^{Nterm} rings appear stable and do not further increase in size. The analysis and discussion of the data has been included into our revision (see **revised Results**, section 'Imaging GSDMD^{Nterm} oligomerization and pore formation', and **new Appendix Fig. S9**).

Figure R2.2, included as new Appendix Fig. S9. Analysis of the surface area of the transmembrane pore formed by GSDMD^{Nterm} oligomers over time. (A-E) Time-lapse AFM topographs showing the assembly and pore formation of GSDMD^{Nterm} oligomers on SLMs made from POPC, DOPG, DOPS, DOPE, and CL (40:20:10:20:10 molar ratio). Time stamps indicate minutes. (F-L) Surface area of transmembrane pores measured in time-lapse AFM topographs. (F) was taken from (A), (G) from (B), (J) from (C), (K) from (D) and (L) from (E). The full color range of the topographs corresponds to a vertical scale of 7.4 nm. Scale bars, 10 nm. Topographs were extracted from time-lapse topographs shown in **Fig. 4 and **Appendix Figs. S7** and **S8**.**

The reviewer further asked whether the pores formed by arc- and slit-shaped oligomers are smaller than those formed by ring-shaped oligomers. We have analyzed the mean size (area) of the transmembrane pores (**Fig. R2.3**). Whereas arc- and slit-shaped oligomers form pores ranging from ≈ 20 to 180 nm^2 (with $54.1 \pm 2.8 \text{ nm}^2$, mean \pm SE, $n = 265$), the pore size formed by ring-shaped oligomers range from ≈ 30 to 750 nm^2 (with $158.1 \pm 4.3 \text{ nm}^2$, mean \pm SE, $n = 179$). We have included this information in the manuscript (see **revised Results**, section 'Sub-nanometer topographs of GSDMD^{Nterm} oligomers' and **revised Fig. 5**).

Figure R2.3. Distribution of the area of transmembrane pores formed by (A) arc- and slit-shaped and (B) ring-shaped GSDMD^{Nterm} oligomers. Thin black lines are Gaussian fits used to determine the mean \pm SE of the data. n gives the numbers of pores analyzed. The data was taken from high-resolution AFM topographs such as shown in **revised Fig. 5**.

The reviewer further asked whether increasing the concentration of GSDMD affects all these (above mentioned) parameters. Conducting the experiments at much higher GSDMD^{Nterm} concentrations than applied in our manuscript results in a too fast coverage of the membrane with slit-, arc- and ring-like oligomers. This crowding of the membrane makes it too difficult to follow the assembly, fusion and pore formation of individual oligomers by time-lapse AFM. We have thus conducted time-lapse experiments at lower GSDMD^{Nterm} concentrations (**Fig. R2.4**). The experiments confirm that GSDMD^{Nterm} assembles into slit-, arc- and ring-like oligomers and that slit- and arc-like oligomers can fuse into ring-like oligomers.

Figure R2.4. Time-lapse topographs showing GSDMD^{Nterm} oligomerization and pore formation at lower GSDMD concentration compared to the concentration used in Fig. 4. A defect-free SLM made from POPC, DOPG, DOPS, DOPE, and CL (40:20:10:20:10 molar ratio) was incubated with 0.4 μ M GSDMD and 0.1 μ M caspase-1 in buffer solution at 37°C. Recorded at different time points of the incubation (indicated by time stamps in min) the time-lapse AFM topographs follow the progress of GSDMD^{Nterm} binding and assembly to the SLM. Arrows indicate the formation and fusion of GSDMD^{Nterm} oligomers through the addition of GSDMD^{Nterm} monomers. The full color range of the topographs corresponds to a vertical scale of 7 nm. Scale bar, 100 nm. Time-lapse FD-based AFM topographs were recorded in buffer solution at 37°C as described (**Materials and Methods**).

The reviewer also asked to analyze the smallest size of slit and arc that can form a pore and more interestingly, given the high quality of the resolution imaging, what is the minimum number of GSDMD molecules that can be associated with a membrane pore. Thank you for bringing up this interesting question. The smallest pore size (area) can be extracted from the pore size distributions given in **Fig. R2.3**. It appears that the smallest pore size we could detect in our study was $\approx 20 \text{ nm}^2$. We have also analyzed the minimum, maximum and average number of GSDMD^{Nterm} molecules assembling the arc-, slit- and ring-shaped oligomers forming transmembrane pores (see **revised Fig. 5**). In average, arc-like oligomers were assembled from 16.1 ± 0.4 (mean \pm SE, $n = 26$) subunits, slit-like oligomers from 20.7 ± 0.4 ($n = 46$) subunits and ring-like oligomers from 30.2 ± 0.2 ($n = 49$) subunits (**revised Fig. 5D**). We have included this information in the manuscript

(see **revised Results**, section ‘Sub-nanometer topographs of GSDMD^{Nterm} oligomers’ and **revised Fig. 5**).

Reviewer #2: *Based on the relatively wide distribution of the ring diameter, the authors claim that GSDMD pore formation is a "structurally rather flexible process" (pag 6). This can be a result of a dynamic assembling process where the pore grows over time before reaching a steady state. It would be interesting to explore this hypothesis by, again, looking at the pore size over time, also in the context of understanding the nature of GSDMD pores (only protein or lipid-protein based). Still, for a better understanding of the kinetics mechanism, it would be useful to plot in a graph how the number of slit/arcs and rings change over time. According to what the authors observe (pag 8) slit/arcs should reach a plateau while rings should constantly increase.*

Authors: To better understand the dynamic assembly process of the GSDMD pores, the reviewer asks to look at the pore size over time. We kindly refer to our answer given above and in which we analyzed the size (area) of the transmembrane pore formed by GSDMD^{Nterm} oligomers over time (**Fig. R2.2**, included as **new Appendix Fig. S9**). The analysis and the discussion hereof have been included in the revised manuscript.

The reviewer also suggests to plot a graph showing the number of slits/arcs and rings changing over time. In our time-lapse AFM experiments we imaged the supported lipid membranes (SLMs) while incubating them with a limited amount of GSDMD and caspase-1 at 37°C. Throughout recording the time-lapse movie, new GSDMD^{Nterm} could insert and assemble oligomers. The graph plotting the number of arcs, slits and rings over time show that the number and arcs and slits changes over time approaching a lower plateau whereas the number of rings increases with time approaching a higher plateau (**Fig. R2.1**). The new figure has been included into the Appendix and the data highlighting the flexible process of the oligomeric assembly have been discussed in the revision (see **revised Results**, section ‘Imaging GSDMD^{Nterm} oligomerization and pore formation’, **new Appendix Fig. S10**, and **revised Discussion**).

Reviewer #2: *Another important aspect is the first part of the manuscript regarding the analysis of lipid composition. The authors should specify why the choice of the POPC, DOPG, DOPS, DOPE, and CL (40:20:10:20:10 w/w) lipid composition in their experiments. GSDMD has been shown to form pores in the inner leaflet of the plasma membrane that are responsible for Pyroptosis execution. The authors should analyze GSDMD pore activity in a lipid composition mimicking this leaflet, which by the way, contains about 30% cholesterol. In this sense, the authors should explain how come cholesterol inhibits pore formation and how can this be put into perspective given the presence of cholesterol in the physiological target membrane of this protein.*

Authors: The reviewer asks to analyze the GSDMD assembly and pore formation in a lipid composition mimicking the inner leaflet of the plasma membrane, which contains $\approx 30\%$ cholesterol. We followed the advice of the reviewer and conducted additional experiments in lipid compositions more closely mimicking the plasma membrane inner leaflet. Since plasma membranes display an asymmetric lipid distribution with PS and PE enriched in the cytosolic leaflet (Devaux & Morris, 2004; van Meer *et al*, 2008), we first used a three-component mixture of POPS, DOPE and POPC (35:25:40 molar ratio) as a model of the inner leaflet of the plasma membrane and investigated GSDMD assembly and pore formation on SLMs made from this lipid composition (**Fig. R2.5**). After this, since phosphoinositides are known to be present in mammalian cell membranes, and since PI(4,5)P2 is a marker of the cytoplasmic leaflet of the plasma membrane and a physiological relevant target of GSDMD, we characterized the effect of PI(4,5)P2 on the binding and assembly of GSDMD^{Nterm}. For comparison, we have also investigated possible effects of phosphoinositol (POPI). Therefore, we have assembled SLMs from POPS, DOPE and POPC (35:25:40 molar ratio), from POPS, DOPE and POPI (35:25:40 molar ratio) and from POPS, DOPE and PI(4,5)P2 (35:25:40 molar ratio). As described for the other experiments, each of the SLMs was incubated with 0.5 μM GSDMD and 0.1 μM caspase-1 for 60 min at 37°C. The AFM topographs showed that GSDMD^{Nterm} also bound to SLMs made from POPS, DOPE and POPC, where it assembled arc- slit- and ring-like oligomers. Again, the oligomers could form transmembrane pores. However, we could not observe GSDMD^{Nterm} binding to SLMs made from POPS, DOPE and POPI. Instead, GSDMD^{Nterm} bound much more frequently to SLMs made from POPS, DOPE and PI(4,5)P2 compared to SLMs made from POPS, DOPE and POPC. This indicates that PI(4,5)P2 promotes the binding and oligomerization of GSDMD^{Nterm}. The data showed that the presence of PI(4,5)P2 had no influence on the diameter of ring-like oligomers. However, because the occurrence of arc- and slit-like oligomers reduced and of ring-like oligomers increased one may thus speculate that PI(4,5)P2 promotes the assembly of ring-like oligomers. The new data has now been included in our revised manuscript and discussed (see **new Fig. 2, new Results**, section ‘Opposing roles of phosphoinositide and cholesterol’, and **revised Discussion**).

Figure R2.5, included as new Fig. 2. Effect of phosphoinositide on the assembly of GSDMD^{Nterm} oligomers and pores. (A-C) AFM topographs showing GSDMD^{Nterm} oligomers formed on SLMs made from (A) POPS, DOPE and POPC (35:25:40 molar ratio), (B) POPS, DOPE and POPI (35:25:40 molar ratio), and (C) POPS, DOPE and PI(4,5)P₂ (35:25:40 molar ratio). (D and E) Topographs showing arc-, slit- and ring-like GSDMD^{Nterm} oligomers formed on SLMs assembled from (D) POPS, DOPE and POPC (35:25:40 molar ratio) and (E) POPS, DOPE and PI(4,5)P₂ (35:25:40 molar ratio). (F-H) Height profiles of GSDMD^{Nterm} oligomers measured along the red lines indicated in the topographs (A-C). Black dashed lines indicate the SLM surface (0 nm height). (I-J) Diameter of GSDMD^{Nterm} ring-shaped oligomers formed on SLMs made from (I) POPS, DOPE and POPC and (J) POPS, DOPE and PI(4,5)P₂. Thin black lines are Gaussian fits determining the mean ± SE values of each distribution. (K) Number of arc-, slit, and ring-shaped GSDMD^{Nterm} oligomers formed on SLMs made from POPS, DOPE and POPC (black bars) or POPS, DOPE and PI(4,5)P₂ (grey bars). Bars present averages and error bars SD. The full color range of the topographs corresponds to a vertical scale of 12 nm. Scale bars, 100 nm (A-C), 20 nm (D and E) and 10 nm (F-H). Averages and errors are given in the text and summarized in **Appendix Table S1**.

The reviewer further suggested to investigate the effect of cholesterol on GSDMD^{Nterm} binding and assembly. Particularly, the reviewer suggested to investigate the effect of cholesterol upon reaching concentrations of $\approx 30\%$ such as described for the inner leaflet of the plasma membrane. We conducted experiments to characterize the effect of cholesterol using a lipid mixture mimicking the cytoplasmic leaflet of the plasma membrane and in the presence of PI(4,5)P₂ (**Fig. R2.6**). For our experiments we assembled SLMs from POPS, POPC, DOPE and PI(4,5)P₂ (35:30:25:10 molar ratio) and from the same lipid mixture supplemented with 15% cholesterol (molar ratio) and with 30% cholesterol. Each of the SLMs was incubated with 0.5 μM GSDMD and 0.1 μM caspase-1 for 60 min at 37°C. The AFM topographs showed that in the presence of 15% cholesterol less

GSDMD^{Nterm} bound to SLMs. Finally, in the presence of 30% cholesterol binding of GSDMD^{Nterm} was largely reduced. The observation that cholesterol reduces the binding of GSDMD^{Nterm} to lipid membranes mimicking the inner leaflet of the plasma membrane, is in line with the observation that cholesterol also reduces the binding of GSDMD^{Nterm} to *E. coli* polar lipid mixtures (Fig. 1C). The new data has been included into our revised manuscript and discussed (see **new Appendix Fig. S6** and **revised Results**, section ‘Opposing roles of phosphoinositide and cholesterol’ and **Discussion**).

Figure R2.6, included as new Appendix Fig. S6. Effect of cholesterol on the assembly of GSDMD^{Nterm} oligomers in phosphoinositide-containing lipid membranes. Overview and high-resolution AFM topographs showing GSDMD^{Nterm} oligomers formed on SLMs made from (A and B) POPS, POPC, DOPE and PI(4,5)P2 (35:30:25:10 molar ratio), (C and D) POPS, POPC, DOPE, PI(4,5)P2 and cholesterol (30:26:21:8:15 molar ratio), and (E and F) POPS, POPC, DOPE, PI(4,5)P2 and cholesterol (24:21:18:7:30 molar ratio). The full color range of the topographs corresponds to a vertical scale of 5 nm. Scale bars, 100 nm (A, C and E) and 50 nm (B, D, and F).

To further increase the number of lipid compositions tested for GSDMD binding and oligomerization, we have assembled SLMs from POPG or POPC and used time-lapse AFM to record the formation of GSDMD oligomers (Fig. R2.7, included as **new Appendix Fig. S11**). The time-lapse AFM topographs again show the assembly and pore formation of arc-, slit- and ring-like oligomers such as observed for the other lipid compositions of the SLMs. All together we have now tested the binding, oligomeric assembly and pore formation of GSDMD to SLMs made from eleven lipid mixtures (conditions are summarized in the **new Appendix Table S3**). Some of these lipid mixtures mimic that of biological membranes others not. Taken together, the experiments show that the lipid composition has an influence

on whether GSDMD can bind and insert, but also show that once inserted GSDMD assembles arc-, slit- and ring-like oligomers and forms pores. Accordingly, we have included and discussed the additional data into the revision of our manuscript (see **revised Results**, section ‘Opposing roles of phosphoinositide and cholesterol’, revised **Discussion**, **new Appendix Fig. S11** and **new Appendix Table S3**).

Figure R2.7, included **new Appendix Fig. S11**. **Time-lapse topographs of GSDMD^{Nterm} oligomerization and pore formation in SLMs made from POPG and POPC.** The first AFM topographs at the left show SLMs made from (A) POPG or (B) POPC before of their incubation with GSDMD and caspase-1. The following AFM topographs show the SLMs incubated with GSDMD and caspase-1 buffer solution at 37°C. In each topograph the time point of the incubation is indicated by the time stamp (in min). Time-lapse FD-based AFM topographs were recorded in buffer solution at 37°C as described (**Materials and Methods**). The full color range of the topographs corresponds to a vertical scale of 10 nm. Scale bars, 50 nm.

Reviewer #2: *The paragraph describing the sub-nanometer topographs of GSDMD is shallow. The authors refer to substructures, but it's not clear to me what they are referring to. They should illustrate with arrows in the figures. What do they mean with lateral distance? Again an illustration would be useful.*

Authors: Thank you. We have extended the paragraph now explaining in detail the sub-nanometer topographs of GSDMD (see **revised Results**, section ‘Sub-nanometer topographs of GSDMD^{Nterm} oligomers’). We have also revised the Figure to better describe the analysis of the GSDMD oligomers imaged in the high-resolution topographs (see **revised Fig. 5**). The Figure has been modified to point out the subunits by arrows and removed the word ‘lateral’ from the text to avoid confusion. We hope that the improved illustrations (high-resolution AFM images and analysis shown in **Fig. 5** and conclusive **Fig. 6**) are better understandable.

Revised Figure 5, formerly Fig. 4. Arc-, slit- and ring-shaped GSDMD^{Nterm} oligomers imaged at sub-nanometer resolution. (A) High-resolution AFM topograph of GSDMD^{Nterm} oligomers assembled on SLMs made from POPC, DOPG, DOPS, DOPE, and CL (40:20:10:20:10 molar ratio). After the SLM has been incubated with GSDMD and caspase-1 for 60 min at 37°C, the FD-based AFM topograph was recorded in buffer solution at room temperature (Methods). (B) and (C) GSDMD^{Nterm} oligomers shown at higher magnification. Arrows indicate adjacent subunits of oligomers, which show an average distance of 2.3 ± 0.3 nm (mean \pm SD, $n = 117$). The full color range of the topographs corresponds to a vertical scale of 10 nm. Scale bars, 20 nm (A) and 10 nm (B and C). (D-F) Average number of subunits per (D) arc-, (E) slit- and (F) ring-shaped GSDMD^{Nterm} oligomer. Thin black lines are Gaussian fits used to determine mean \pm SE of the data, n indicates the number of oligomers measured.

Reviewer #2: Minor aspects: - For clarity, the authors should provide a zoom in picture of a slit, an arc and a ring

Authors: Thank you, we now show several zooms of slit-, arc- and ring-like oligomers in the **revised Figs. 1, 2 and 5.**

Revised Figure 1. Characterization of the assembly of GSDMD^{Nterm} oligomers. (A-C) AFM topographs showing GSDMD^{Nterm} oligomers formed on supported lipid membranes (SLMs) assembled from (A) POPC, DOPG, DOPS, DOPE, and CL (40:20:10:20:10 molar ratio), (B) *E. coli* polar lipid extract, or (C) *E. coli* polar lipid extract and cholesterol (70:30 weight ratio). Scale bars, 50 nm. (D-F) Topographs showing (D) arc-, (E) slit- and (F) ring-like GSDMD^{Nterm} oligomers formed on SLMs assembled from POPC, DOPG, DOPS, DOPE, and CL (40:20:10:20:10 molar ratio), *E. coli* polar lipid extract, or *E. coli* polar lipid extract and cholesterol (70:30 weight ratio). Scale bars, 10 nm. (G-I) Height profiles of GSDMD^{Nterm} oligomers measured along the red lines in the AFM topographs (A-C). Black dashed lines indicate the SLM surface (0 nm height). Scale bars, 10 nm. (J-L) Distribution of GSDMD^{Nterm} ring diameters formed into SLMs made from (J) POPC, DOPG, DOPS, DOPE, and CL, (K) *E. coli* polar lipid extract and (L) *E. coli* polar lipid extract and cholesterol. Thin black lines are Gaussian fits determining the mean \pm SE values given for each distribution. (M) Average height of GSDMD^{Nterm} oligomers protruding from the lipid bilayer. Shown are heights measured for ring- (black) and slit-shaped (grey) oligomers formed in SLMs made from (i) POPC, DOPG, DOPS, DOPE, and CL, (ii) *E. coli* polar lipid extract or (iii) *E. coli* polar lipid extract and cholesterol. Bars present average and error bars SD. The full color range of the topographs corresponds to a vertical scale of 12 nm. Averages and errors are summarized in **Appendix Table S1**.

Reviewer #2: - The title of the first paragraph in the results section is misleading because the authors said that GSDMD oligomerization depends on the lipid composition, however later on in the text they specify that GSDMD assembly is not

sensitive to the lipid composition (pag 6). They should write again the title focusing more on the Cholesterol role rather than lipids in general.

Authors: We apologize for the confusion created. We have corrected the misleading title of the first paragraph.

Reviewer #2: - *On page 5 the authors, based on the low height of the structures protruding the membrane, conclude that GSDMD monomers fully insert into the membrane. This is a quite strong statement not supported by high resolution structural data*

Authors: Thank you, we agree. From our measurements, we cannot unambiguously state that GSDMD fully inserted into the membrane. We have revised the sentence to tone down the statement.

Reviewer #2: - *Fig 2A provides the same information of fig 1A and graph G in fig 2 is the same than graph G in fig 1*

Authors: We apologize for having created confusion. For a direct comparison of the data collected for different caspases, we decided to show GSDMD^{Nterm} oligomers cleaved by caspase-1, -4 and -5. In **Fig. 2A** (now **Fig. 3A**) we show a different AFM topograph of GSDMD^{Nterm} oligomers (but also cleaved by caspase-1) having a slightly higher resolution than the AFM topograph shown in **Fig. 1A**. Similarly, we now clearly referenced that the graph shown in **Fig. 3G** has been taken from **Fig. 1J** and is shown to allow a better comparison of the data.

Reviewer #2: - *Figure s3 does not provide any additional information (see fig 3)*

Authors: We apologize for the confusion created. **Appendix Fig. S3** (now **Appendix Fig. S7**) shows an additional independent series of time-lapse AFM topographs observing the formation of GSDMD rings fusing from arcs and slits. We have revised text and Figure legend to clearly state this issue.

Reviewer #2: - *In the author contribution section some authors are not mentioned*

Authors: Thank you we have corrected the contribution section.

EMBOJ-2017-98321 “Mechanism of membrane pore formation by human Gasdermin-D”

Point-by-point response to the comments of reviewer #3

Reviewer #3: *Mulvihill study the insertion and pore-formation of GSDMD N-termini into membranes by AFM. They find that the structures formed are of different shapes and sizes. The AFM pictures are impressive and of very high quality and resolution. Taken that this is the strong part of the manuscript, the most should be made of this, e.g., the scale of height should be added and interpreted in all pictures. The main criticism is that it is difficult to judge how relevant the data is since a membrane on mica might not allow all conformations the protein might assume in a real membrane.*

Authors: We thank the reviewer for the encouraging and constructive comments which guided us to revise our manuscript. The reviewer asks to provide the scale of the height information to all AFM images. We have revised the figure legends to define the vertical scale (height scale) of every AFM topograph shown.

The reviewer further comments that is difficult to judge how relevant the data is since a membrane on mica might not allow all conformations a protein assumes in a real membrane. We agree that supporting a membrane could indeed restrict the conformations of a membrane protein. However, within the past 20 years we have characterized more than 20 different membrane proteins in native and reconstituted membranes adsorbed to mica and could never observe such an artifact (Bippes & Muller, 2011; Engel & Muller, 2000). In this study, we have used different supported lipid membranes as model systems to investigate the assembly and pore formation of GSDMD and how the lipid composition of the membrane itself can influence the pore formation. To revise our manuscript, we have applied TEM imaging to characterize whether the assembly of GSDMD into arc-, slit- and ring-like oligomers is an artifact caused by the mica supporting the lipid membranes (**Fig. R3.1**). In the control experiments, liposomes suspended in buffer solution were first incubated overnight with GSDMD and catalytic amounts of caspase-1 at 37°C. Afterwards, the liposomes were imaged by TEM. The TEM images show the co-existence of arc-, slit- and ring-like oligomers and thus support our AFM results obtained using supported lipid membranes. The data has been included and discussed in our revision (see revised Results, section GSDMD^{Nterm} oligomerization in liposomes, and new **Appendix Fig. S5**).

Figure R3.1, included as new Appendix Fig. S5. TEM of arc-, slit- and ring-shaped GSDMD^{Nterm} oligomers on liposomes. (A and B) TEM images of liposomes incubated with GSDMD in the presence of caspase-1. Upon adsorption to the TEM grid most of the liposomes fused forming lipid membranes. GSDMD^{Nterm} oligomers inserted into lipid membranes are observed. (C and D) Higher resolution images of arc-, slit- and ring-shaped oligomers formed by GSDMD^{Nterm}. Arc-like oligomers are pointed out by single arrowheads and slit-like oligomers by double arrowheads. Suspended liposomes made from *E. coli* polar lipid extract were incubated over night at 37°C with 5 μ M GSDMD and 1 μ M caspase-1. Samples were negatively stained with uranyl acetate and imaged at 120 kV (**Materials and Methods**). Scale bars, 200 nm (A), 100 nm (B) and 50 nm (C and D).

Reviewer #3: The dimensions of the pore should be quantified not only by diameter but also by the number of GSDMD subunits per pore, which seem possible, at least from the data in figure 4 and is the strong point of this work.

Authors: Thank you. We have now quantified the average number of subunits forming arc-, slit- and ring-like GSDMD oligomers (see revised Fig. 5, formerly Fig. 4). The numbers are now included into the revised manuscript (see revised Results, section 'Sub-nanometer topographs of GSDMD^{Nterm} oligomers', and revised Fig. 5).

Revised Figure 5, formerly Fig. 4. Arc-, slit- and ring-shaped GSDMD^{Nterm} oligomers imaged at sub-nanometer resolution. (A) High-resolution AFM topograph of GSDMD^{Nterm} oligomers assembled on SLMs made from POPC, DOPG, DOPS, DOPE, and CL (40:20:10:20:10 molar ratio). After the SLM has been incubated with GSDMD and caspase-1 for 60 min at 37°C, the FD-based AFM topograph was recorded in buffer solution at room temperature (Methods). (B) and (C) GSDMD^{Nterm} oligomers shown at higher magnification. Arrows indicate adjacent subunits of oligomers, which show an average distance of 2.3 ± 0.3 nm (mean \pm SD, $n = 117$). The full color range of the topographs corresponds to a vertical scale of 10 nm. Scale bars, 20 nm (A) and 10 nm (B and C). (D-F) Average number of subunits per (D) arc-, (E) slit- and (F) ring-shaped GSDMD^{Nterm} oligomer. Thin black lines are Gaussian fits used to determine mean \pm SE of the data, n indicates the number of oligomers measured.

Reviewer #3: More detailed statistics on the shape and diameter of the pores should be presented. Is there a most-common diameter / number of subunits of the ring-shaped pore that is the most stable, or do the pores just keep growing?

Authors: We have now quantified the mean number of subunits forming ring-shaped GSDMD pores (revised Fig. 5F), which is 30.2 ± 0.8 (mean \pm SE, $n = 49$). The same analysis has been done for arc- and slit-like oligomers (revised Fig. 5D,E). We also analyzed the size (area) of the transmembrane pores formed by arc- and slit-shaped oligomers (Fig. R3.2). Whereas arc- and slit-shaped oligomers form pores ranging from ≈ 20 to 180 nm² (with 54.1 ± 2.8 nm²,

mean \pm SE, $n = 265$), the pore size formed by ring-shaped oligomers ranges from ≈ 30 to 750 nm^2 (with $158 \pm 4.3 \text{ nm}^2$, $n = 179$).

Figure R3.2. Area of transmembrane pores formed by (A) arc- and slit-shaped and (B) ring-shaped GSDMD^{Nterm} oligomers. Thin black lines are Gaussian fits used to determine the mean and SE of the data. n gives the numbers of pores analyzed. The data was taken from high-resolution AFM topographs such as shown in Fig. 5.

The reviewer further asks whether the ring-shaped pores keep on growing. We have analyzed our time-lapse AFM images to answer this question (Fig. R3.3, included as new Appendix Fig. S9). The data show that the size of the pores formed by arc- and slit-shaped GSDMD^{Nterm} oligomers can increase with time. The analysis, however, also shows that GSDMD^{Nterm} rings, once formed are kinetically stable (plateaus reached in Fig. R3.3F-L). Only very rarely we observed the rings to open to further increase in size. We have included this information into our revised manuscript (see revised Results, sections 'Imaging GSDMD^{Nterm} oligomerization and pore formation' and 'Sub-nanometer topographs of GSDMD^{Nterm} oligomers', revised Fig. 5, and new Appendix Fig. S9).

Figure R3.3, included as new Appendix Fig. S9. Analysis of the surface area of the transmembrane pore formed by GSDMD^{Nterm} oligomers over time. (A-E) Time-lapse AFM topographs showing the assembly and pore formation of GSDMD^{Nterm} oligomers on SLMs made from POPC, DOPG, DOPS, DOPE, and CL (40:20:10:20:10 molar ratio). Time stamps indicate minutes. (F-L) Surface area of transmembrane pores measured in time-lapse AFM topographs. (F)

was taken from (A), (G) from (B), (J) from (C), (K) from (D) and (L) from (E). The full color range of the topographs corresponds to a vertical scale of 7.4 nm. Scale bars, 10 nm. Topographs were extracted from time-lapse topographs shown in **Fig. 4** and **Appendix Figs. S7** and **S8**.

Reviewer #3: *We would have expected more orderly structures. Using this experimental system, do other, better studied pore-forming proteins also assume such diverse pore shapes? How representative is this of the physiological structure?*

Authors: The reviewer asks whether other pore-forming proteins also assume such diverse pore shapes. Yes, other pore-forming proteins including cholesterol-dependent cytolysins or membrane attack complex/perforin (MACPF) also assemble arc-, slit- and ring-like oligomers all of which giving rise to diverse pore shapes (Hodel *et al*, 2016; Leung *et al*, 2014; Leung *et al*, 2017; Mulvihill *et al*, 2015; Sonnen *et al*, 2014; van Pee *et al*, 2016). How these proteins assemble and form transmembrane pores can differ substantially. The surprising finding of our study is that GSDMD^{Nterm} oligomers show very similar shapes as other pore forming proteins/toxins produced by bacteria. However, the growing mechanism of GSDMD^{Nterm} into diverse oligomeric and pore shapes appears to be less common.

The reviewer further asked how representative the observed variety of GSDMD^{Nterm} oligomers is for the physiological structure. Recent publications on GSDMD^{Nterm} oligomers have already shown their ring and pore size to vary (Ding *et al*, 2016; Sborgi *et al*, 2016). It is thus not surprising that we also observe such variety. However, the structural variation of the arc-, slit- and ring-like GSDMD^{Nterm} oligomers, the dynamic fusion of arcs and slits into ring-like oligomers and the observation that each of the oligomeric forms can form transmembrane pores is new.

We have revised our manuscript to briefly elaborate on both issues (see **revised Discussion**).

Reviewer #3: *Why do they only test the E. coli membrane with and without cholesterol, not also the synthetic membrane?*

Authors: The composition of some lipid membranes used in this work mimic that of an *E. coli* membrane. However, in our revision we have additionally tested several other lipid mixtures, which more closely mimic those of mammalian cells (**Figs. R3.4, R3.5** and **R3.6**, now included as **new Appendix Figs. S6** and **S11** and **new Fig. 2**). The new experiments particularly tested the role of phosphoinositides and cholesterol (**Fig. 2** and **new Appendix Fig. S6**). In summary, we now have tested eleven different lipid mixtures for GSDMD binding, oligomerization and pore formation (summarized in **new Appendix Table S3**). Taken together, the experiments show that if GSDMD^{Nterm} binds and inserts, it forms arc-, slit- and ring-like oligomeric structures and transmembrane pores. The

additional experiments suggest that the binding and insertion of GSDMD^{Nterm} depend on the lipid composition of the membrane, but that the oligomeric assembly and pore formation rather appears to be a property intrinsic to GSDMD^{Nterm}. We have revised our manuscript to include the additional experimental data and to discuss these findings more clearly (see **revised Results and Discussion, new Fig. 3, new Appendix Figs. S6 and S11, and new Appendix Table S3**).

Figure R3.4, included as new Appendix Fig. S11. Time-lapse topographs of GSDMD^{Nterm} oligomerization and pore formation in SLMs made from POPG and POPC. The first AFM topographs at the left show SLMs made from (A) POPG or (B) POPC before of their incubation with GSDMD and caspase-1. The following AFM topographs show the SLMs incubated with GSDMD and caspase-1 buffer solution at 37°C. In each topograph the time point of the incubation is indicated by the time stamp (in min). The time-lapse FD-based AFM topographs were recorded in buffer solution at 37°C as described (**Materials and Methods**). The full color range of the topographs corresponds to a vertical scale of 10 nm. Scale bars, 50 nm.

Figure R3.5, included as new Fig. 2. Effect of phosphoinositide on the assembly of GSDMD^{Nterm} oligomers and pores. (A-C) AFM topographs showing GSDMD^{Nterm} oligomers formed on SLMs made from (A) POPS, DOPE and POPC (35:25:40 molar ratio), (B) POPS, DOPE and POPI (35:25:40 molar ratio), (C) POPS, DOPE and POPI (35:25:40 molar ratio). (D-E) High-magnification AFM topographs of GSDMD^{Nterm} oligomers. (F-H) Height profiles (Height in nm vs Diameter in nm) for oligomers 1-6. (I-J) Histograms of relative count vs Diameter (nm) for oligomers 1-6. (K) Bar chart of relative count for Arcs, Slits, and Rings for DOPE:POPS:POPC and DOPE:POPS:PI(4,5)P₂.

ratio), and (C) POPS, DOPE and PI(4,5)P₂ (35:25:40 molar ratio). (D and E) Topographs showing arc-, slit- and ring-like GSDMD^{Nterm} oligomers formed on SLMs assembled from (D) POPS, DOPE and POPC (35:25:40 molar ratio) and (E) POPS, DOPE and PI(4,5)P₂ (35:25:40 molar ratio). (F-H) Height profiles of GSDMD^{Nterm} oligomers measured along the red lines indicated in the topographs (A-C). Black dashed lines indicate the SLM surface (0 nm height). (I-J) Diameter of GSDMD^{Nterm} ring-shaped oligomers formed on SLMs made from (I) POPS, DOPE and POPC and (J) POPS, DOPE and PI(4,5)P₂. Thin black lines are Gaussian fits determining the mean \pm SE values of each distribution. (K) Number of arc-, slit, and ring-shaped GSDMD^{Nterm} oligomers formed on SLMs made from POPS, DOPE and POPC (black bars) or POPS, DOPE and PI(4,5)P₂ (grey bars). Bars and values present averages and error bars SD. The full color range of the topographs corresponds to a vertical scale of 12 nm. Scale bars, 100 nm (A-C), 20 nm (D and E) and 10 nm (F-H). Averages and errors are given in the text and summarized in **Appendix Table S1**.

Figure R3.6, included as new Appendix Fig. S6. Effect of cholesterol on the assembly of GSDMD^{Nterm} oligomers in phosphoinositide-containing lipid membranes. Overview and high-resolution AFM topographs showing GSDMD^{Nterm} oligomers formed on SLMs made from (A and B) POPS, POPC, DOPE and PI(4,5)P₂ (35:30:25:10 molar ratio), (C and D) POPS, POPC, DOPE, PI(4,5)P₂ and cholesterol (30:26:21:8:15 molar ratio), and (E and F) POPS, POPC, DOPE, PI(4,5)P₂ and cholesterol (24:21:18:7:30 molar ratio). The full color range of the topographs corresponds to a vertical scale of 5 nm. Scale bars, 100 nm (A, C and E) and 50 nm (B, D, and F).

Reviewer #3: What happens if GSDMD and Casp1 without the membrane being around and later added to the membrane? Is it important that the GSDMD N-term is "fresh"?

Authors: To answer this question we pre-incubated GSDMD and caspase-1 for 60 min at 37°C in the absence of lipid membranes. This pre-incubated sample was then incubated on SLMs made from *E. coli* polar lipid extract for 60 min at 37°C.

After the incubation time, the SLMs were rinsed with buffer solution and imaged at room temperature by AFM. The AFM topographs show that GSDMD^{Nterm} could still form a few oligomers into the lipid membranes (**Fig. R3.7**). However, instead of oligomers we mostly observed larger aggregates on the SLMs instead of oligomers. Thus, GSDMD^{Nterm} has to be 'fresh'.

Figure R3.7. Pre-incubation of human GSDMD with caspase-1 in absence of SLM reduces the pore-formation potential of GSDMD^{Nterm}. (A) AFM topograph of SLM made from *E. coli* polar lipid extract. (B and C) AFM topograph of GSDMD (0.5 μ M) and caspase-1 (0.1 μ M) incubated in buffer solution (50 mM NaCl, 100 mM Hepes, 5 mM TCEP, pH 7.4) for 60 min at 37°C in tube and then incubated to a SLM made from *E. coli* polar lipid extract for 60 min at 37°C. Scale bars, 100 nm. Full color range of topographs corresponds to a vertical scale of 100 nm.

Reviewer #3: *Figure 2: do the different caspases cleave GSDMD at different sites? If not, the result is highly expected. The authors provide no indication on why this was an interesting question for them to test.*

Authors: Thank you for pointing this out. It has been shown that the three pro-inflammatory caspases (-1, -4 and -5) tested in our study are activated by different pathways (canonical and non-canonical inflammasomes) (Aglietti *et al*, 2016; Kayagaki *et al*, 2015; Shi *et al*, 2015; Vigano *et al*, 2015). Apparently, however, they cleave and activate other substrates, including interleukin-18, with different efficacy (Ghayur *et al*, 1997). Thus, it is particularly interesting to see that incubation of the SLMs with each of the caspases tested in our study causes GSDMD to form the same oligomers and transmembrane pores within the same incubation time. While common cleavage sites can indeed be expected, it is reassuring to see that the resulting pores are indistinguishable and that the different caspases causes similar amounts of GSDMD^{Nterm} oligomers to insert within the same incubation time. We have revised our manuscript to describe this point more clearly.

Reviewer #3: *The authors claim that "the relatively wide distribution of the diameter of ring-shaped oligomers and the coexistence of arc-, slit-, and ring-shaped oligomers indicates that the assembly of transmembrane pores by GSDMD^{Nterm} is a structurally rather flexible process." - How can they exclude that*

these phenomena are not a result of their experimental settings and that the natural pore forms in a more orderly fashion?

Authors: AFM can record images of single transmembrane proteins at (sub-)nanometer resolution and most importantly in the native, unperturbed state. In the past two decades, we and others have shown on the example of many different oligomeric membrane proteins that AFM provides highly reproducible images of these oligomers (Dufrene *et al*, 2017; Seelert *et al*, 2000). If ring-like oligomers show no structural flexibility we observe their diameter to distribute quite narrowly (Muller *et al*, 2001; Seelert *et al*, 2003; Seelert *et al*, 2000; Stahlberg *et al*, 2001). However, if the oligomers show higher structural flexibility, such as induced by mutations, their diameters show wider distributions (Pogoryelov *et al*, 2012). We can thus exclude that the AFM imaging process itself influences the diameter of the oligomers. This can also be seen on time-lapse AFM images repetitively imaging the same GSDMD^{Nterm} oligomeric rings, which do not change size (**Fig. 4** and **Appendix Figs. S7** and **S8**).

It may be also argued that the mica supporting the lipid membrane influences the assembly of the GSDMD^{Nterm} oligomers. However, it has been shown in several previous works that supporting a lipid membrane by mica does not influence the self-insertion and assembly of a variety of pore forming oligomers (Hodel *et al*, 2016; Leung *et al*, 2014; Mulvihill *et al*, 2015; van Pee *et al*, 2016). Nevertheless, in our revision, we also used TEM to image GSDMD^{Nterm} oligomers bound to suspended liposomes (**Fig. R3.1**, included as new **Appendix Fig. S5**). The TEM images show that GSDMD^{Nterm} assembles arc-, slit- and ring-like oligomers in suspended liposomes similarly to those observed in lipid membranes supported by mica. In contrast, incubating GSDMD and caspase-1 on mica or in buffer solution in the absence of lipids shows no oligomeric structures (**Figs. R3.7** and **R3.8**). We can thus exclude such artifact.

Figure R3.7, included as new Appendix Fig. S3. AFM of human GSDMD incubated with caspase-1 in the absence of a lipid membrane on mica or incubated the absence of caspase-1 on a SLM. (A) AFM topograph of a freshly cleaved atomically flat mica support imaged in buffer solution (50 mM NaCl, 100 mM Hepes, 5 mM TCEP, pH 7.4). (B) Topograph of GSDMD (0.5 μ M) and caspase-1 (0.1 μ M) incubated in buffer solution (50 mM NaCl, 100 mM Hepes, 5 mM TCEP, pH 7.4) for 60 min at 37°C on the mica support. (C) Topograph of an SLM made from *E. coli* polar lipid extract and incubated with GSDMD (0.5 μ M) in the absence of caspase in buffer solution (50 mM NaCl, 100 mM Hepes, 5 mM TCEP, pH 7.4) for 60 min at 37°C. The full color range of the topographs corresponds to a vertical scale of 4.5 nm. Scale bars, 100 nm.

Figure R3.8, included as new Appendix Fig. S4. TEM of liposomes, of liposomes incubated with human GSDMD in the absence of caspase-1, and of human GSDMD incubated with caspase-1. (A) TEM image of liposomes made from *E. coli* polar lipid extract. During the adsorption onto the TEM grid liposomes fuse with each other forming lipid membranes. Liposomes made from other lipids used in this work appeared very similar (not shown here). **(B)** TEM image of liposomes made from *E. coli* polar lipid extract and incubated with 5 μM GSDMD in the absence of caspase-1. None of the lipid membranes imaged ($n > 50$) showed arc-, slit- or ring-like structures. **(C)** TEM image of GSDMD (5 μM) incubated with caspase-1 (0.1 μM) overnight at 37°C in the absence of liposomes. The image shows no arc-, slit- or ring-like structures. Scale bars, 500 nm (A) and 100 nm (B and C).

In addition, as replied further above to the reviewer, recent publications on GSDMD^{Nterm} oligomers showed their ring and pore size to vary (Ding *et al*, 2016; Sborgi *et al*, 2016). It is thus not surprising that we also observe such variety of the oligomeric forms. Taken together, we have now investigated the GSDMD^{Nterm} oligomer formation on membranes assembled from eleven different lipid compositions (**new Appendix Table S3**). The experiments show that the composition of the lipid membrane influences on whether GSDMD^{Nterm} can bind, insert and assemble oligomers. However, our experiments also show that the arc-, slit- and ring-like oligomers assembled by GSDMD^{Nterm} appear to be independent of the lipid composition and thus appear to reflect an intrinsic property of GSDMD^{Nterm}. We have revised our manuscript to include the additional experimental data and control experiments and to explain this issue in detail (see **revised Results and Discussion, new Fig. 3, new Appendix Figs. S6, S11 and new Appendix Table S3**).

Reviewer #3: *The protrusion (Z-Axis) should be depicted separately and quantified not only in Figure 1.*

Authors: Thank you. We now specify the vertical scale for each figure showing AFM topographs.

Reviewer #3: *The use of supported lipid membranes on mica (a solid carrier) could be a problem if the GSDMD pore involves GSDMD molecules protruding from the membrane on the outer side. The authors should comment how this is dealt with in the field for other pore-forming proteins*

Authors: Thank you, we have addressed this concern of the reviewer by answering his/her questions further above and conducted control experiments (AFM and TEM imaging) showing that the solid carrier (mica) does not influence the GSDMD^{Nterm} assembly (**Figs. R3.1, R3.4, and R3.5**, which were included as **new Appendix Figs. S3, S4 and S5**). Notably, we and others have shown in earlier experiments that membrane proteins in supported lipid membranes can freely diffuse (Muller *et al*, 2003), which is also revealed in our time-lapse AFM images showing that GSDMD^{Nterm} oligomers change positions between the images (**Fig. 4 and Appendix Figs. S7 and S8**). It has been shown that a lipid membrane supported by mica is separated from the mica by ≈ 1 nm, which corresponds to a few water layers allowing the proteins diffusing (Tanaka & Sackmann, 2005).

Reviewer #3: *In Figure 4, the central part in some pores is obviously protruding higher than the surrounding ring. Why is this not quantified? Could this be a consequence of a secondary structure trying to form (that might be able to stabilize the ring and stop its growth)? The presence of GSDMD in the middle of the pore is even depicted in Figure 5 (final stage), but not sufficiently discussed. However, the model shown does not properly represent the obtained data because the different height of the protein in the center of the pore in the terminal stage is not shown.*

Authors: Thank you. We agree that this central part of the ring-shaped GSDMD^{Nterm} oligomers plays a central role. We have revised the conclusive **Fig. 6** (formerly **Fig. 5**) to describe this process better. As observed in the AFM topographs, **Fig. 6** now depicts arcs and slits assembling into larger oligomers. Occasionally, the pores of GSDMD^{Nterm} ring-like oligomers contained some material, which presumably resulted from lipids and GSDMD^{Nterm} molecules that could not be integrated during formation of the ring. Our time-lapse AFM topographs show that this material can exit from the pore lumen, leaving an open transmembrane pore behind. The exit of the material trapped inside the pore is preceded by its height increasing (such as observed in the time-lapse topographs shown in **Fig. 4**, when looking at the oligomers recorded in the upper third of the topographs, frames taken between 55 and 91 min (red arrows)). We have now revised the manuscript to discuss this issue more clearly (see **revised Results, Discussion and Figs. 5 and 6**).

Revised Figure 6, formerly Fig. 5. Model of GSDMD^{Nterm} oligomerization and pore formation. (i) Human GSDMD is cleaved by inflammatory caspase-1, -4 or -5. (ii) Afterwards, the cleaved N-terminal GSDMD domain GSDMD^{Nterm} binds to the lipid membrane where it (iii) oligomerizes into slit-, arc-, and ring-shaped structures. (iv) Arc- and slit-shaped oligomers can fuse into larger ring-shaped oligomers. Alternatively, oligomers may grow by assembling GSDMD^{Nterm} molecules to their free ends. During this process, GSDMD^{Nterm} oligomers do not significantly change height above the lipid membrane, suggesting the absence of large conformational changes. While each of the oligomeric species can form lytic transmembrane pores, the slit-, and arc-shaped structures fuse into ring-shaped structures in a time-dependent manner. This suggests that ring-shaped oligomers are thermodynamically more stable than the other oligomeric species. (v) Upon fusing into larger oligomers GSDMD^{Nterm} and lipids may remain trapped inside the oligomer. (vi) With time, the trapped material loosens from the membrane, increases height and exits to the solution.

Reviewer #3: *If the authors want to reach the broad readership of the EMBO Journal, they should explain why they think their synthetic membrane is physiologically relevant and how it is composed. The abbreviations are not explained anywhere.*

Authors: We thank the reviewer. In our paper, we have shown that GSDMD^{Nterm} inserts and assembles pore-forming arc-, slit- and ring-shaped oligomers in different model membranes. It is standard to the field that the formation of oligomeric pores is structurally characterized using synthetic lipid membranes. Following the reviewer's suggestion and to further strengthen our manuscript, we have included additional experiments showing that GSDMD^{Nterm} forms arc-, slit- and ring-like transmembrane pores in a broad variety of different lipid membranes (**new Fig. 3** and **new Appendix Figs. S6** and **S11**). All together we have now tested eleven different lipid membrane compositions (summarized in **Appendix Table S3**). In our new experiments, we have for example investigated the effect on GSDMD^{Nterm} binding and oligomerization of phosphatidylinositide and cholesterol in a lipid mixture mimicking the cytoplasmic leaflet of the plasma membrane, the physiological target of GSDMD. The reasons why we decided to prepare our own

lipid mixtures instead of using commercially available lipid extracts from mammalian sources are the following:

(i) Lipid extracts are often “dirty”. They are chloroform extracts of the respective tissue and contain not only lipids but also several other hydrophobic molecules that like to partition in the organic phase. This dirt tends to contaminate the AFM tip affecting the imaging quality and the resolution of the AFM topographs.

(ii) Preparing our own lipid mixtures allows us to address the effect of individual lipids one at a time and to take into account the asymmetric distribution of lipids in biological membranes.

For all lipid membranes, we have observed that if GSDMD^{Nterm} binds to the lipid membrane it can also assemble slit-, arc-, and ring-like oligomers and transmembrane pores. We can thus conclude that the binding and insertion of GSDMD^{Nterm} into lipid membrane depends on the lipid composition and that the oligomeric assembly described in our paper is a general property of GSDMD. We have included the new data showing the generality for the GSDMD^{Nterm} binding, assembly and pore-forming mechanism and discussed it accordingly in our revised manuscript (see **revised Results, revised Discussion, new Fig. 3 and new Appendix Figs. S6 and S11**).

Furthermore, we have carefully revised our manuscript to introduce all abbreviations used, including those of the lipids.

Reviewer #3: *Why do the authors keep repeating the composition of their membrane including the ratios throughout the manuscript if it is always the same? It is enough the just say "SLM" after it was first mentioned, unless it differs.*

Authors: We have reduced redundancy of the descriptions as far as possible. However, in particular in the revised version of the manuscript, where we now characterize a larger variety of eleven different lipid mixtures, we feel that it is essential to keep on specifying repetitively which lipid membrane composition was characterized in which experiment. The **new Appendix Table S3** gives an overview of the lipid compositions characterized.

Reviewer #3: *In the first sentence, "been" is missing, or it should be "was" instead of "has".*

Authors: Thank you. The sentence has been corrected.

References

- Aglietti RA, Estevez A, Gupta A, Ramirez MG, Liu PS, Kayagaki N, Ciferri C, Dixit VM, Dueber EC (2016) GsdmD p30 elicited by caspase-11 during pyroptosis forms pores in membranes. *Proc Natl Acad Sci U S A* 113: 7858-63
- Bippes CA, Muller DJ (2011) High-resolution atomic force microscopy and spectroscopy of native membrane proteins. *Rep Prog Phys* 74: 086601
- Devaux PF, Morris R (2004) Transmembrane asymmetry and lateral domains in biological membranes. *Traffic* 5: 241-6
- Ding J, Wang K, Liu W, She Y, Sun Q, Shi J, Sun H, Wang DC, Shao F (2016) Pore-forming activity and structural autoinhibition of the gasdermin family. *Nature* 535: 111-6
- Dufrene YF, Ando T, Garcia R, Alsteens D, Martinez-Martin D, Engel A, Gerber C, Muller DJ (2017) Imaging modes of atomic force microscopy for application in molecular and cell biology. *Nat Nanotechnol* 12: 295-307
- Engel A, Muller DJ (2000) Observing single biomolecules at work with the atomic force microscope. *Nat Struct Biol* 7: 715-8
- Ghayur T, Banerjee S, Hugunin M, Butler D, Herzog L, Carter A, Quintal L, Sekut L, Talanian R, Paskind M, Wong W, Kamen R, Tracey D, Allen H (1997) Caspase-1 processes IFN-gamma-inducing factor and regulates LPS-induced IFN-gamma production. *Nature* 386: 619-23
- Hodel AW, Leung C, Dudkina NV, Saibil HR, Hoogenboom BW (2016) Atomic force microscopy of membrane pore formation by cholesterol dependent cytolysins. *Curr Opin Struct Biol* 39: 8-15
- Kayagaki N, Stowe IB, Lee BL, O'Rourke K, Anderson K, Warming S, Cuellar T, Haley B, Roose-Girma M, Phung QT, Liu PS, Lill JR, Li H, Wu J, Kummerfeld S, Zhang J, Lee WP, Snipas SJ, Salvesen GS, Morris LX et al. (2015) Caspase-11 cleaves gasdermin D for non-canonical inflammasome signalling. *Nature* 526: 666-71
- Leung C, Dudkina NV, Lukyanova N, Hodel AW, Farabella I, Pandurangan AP, Jahan N, Pires Damaso M, Osmanovic D, Reboul CF, Dunstone MA, Andrew PW, Lonnen R, Topf M, Saibil HR, Hoogenboom BW (2014) Stepwise visualization of membrane pore formation by sulilysin, a bacterial cholesterol-dependent cytolysin. *Elife* 3: e04247
- Leung C, Hodel AW, Brennan AJ, Lukyanova N, Tran S, House CM, Kondos SC, Whisstock JC, Dunstone MA, Trapani JA, Voskoboinik I, Saibil HR, Hoogenboom BW (2017) Real-time visualization of perforin nanopore assembly. *Nat Nanotechnol* 12: 467-473
- Muller DJ, Dencher NA, Meier T, Dimroth P, Suda K, Stahlberg H, Engel A, Seelert H, Matthey U (2001) ATP synthase: constrained stoichiometry of the transmembrane rotor. *FEBS Lett* 504: 219-22
- Muller DJ, Engel A, Matthey U, Meier T, Dimroth P, Suda K (2003) Observing membrane protein diffusion at subnanometer resolution. *J Mol Biol* 327: 925-30
- Mulvihill E, van Pee K, Mari SA, Muller DJ, Yildiz O (2015) Directly Observing the Lipid-Dependent Self-Assembly and Pore-Forming Mechanism of the Cytolytic Toxin Listeriolysin O. *Nano Lett* 15: 6965-73
- Pogoryelov D, Klyszejko AL, Krasnoselska GO, Heller EM, Leone V, Langer JD, Vonck J, Muller DJ, Faraldo-Gomez JD, Meier T (2012) Engineering rotor ring stoichiometries in the ATP synthase. *Proc Natl Acad Sci U S A* 109: E1599-608

- Sborgi L, Ruhl S, Mulvihill E, Pipercevic J, Heilig R, Stahlberg H, Farady CJ, Muller DJ, Broz P, Hiller S (2016) GSDMD membrane pore formation constitutes the mechanism of pyroptotic cell death. *EMBO J* 35: 1766-78
- Seelert H, Dencher NA, Muller DJ (2003) Fourteen protomers compose the oligomer III of the proton-rotor in spinach chloroplast ATP synthase. *J Mol Biol* 333: 337-44
- Seelert H, Poetsch A, Dencher NA, Engel A, Stahlberg H, Müller DJ (2000) Proton powered turbine of a plant motor. *Nature* 405: 418-419
- Shi J, Zhao Y, Wang K, Shi X, Wang Y, Huang H, Zhuang Y, Cai T, Wang F, Shao F (2015) Cleavage of GSDMD by inflammatory caspases determines pyroptotic cell death. *Nature* 526: 660-5
- Sonnen AF, Plitzko JM, Gilbert RJ (2014) Incomplete pneumolysin oligomers form membrane pores. *Open Biol* 4: 140044
- Stahlberg H, Müller DJ, Suda K, Fotiadis D, Engel A, Matthey U, Meier T, Dimroth P (2001) Bacterial ATP synthase has an undecameric rotor. *EMBO Reports* 21: 1-5
- Tanaka M, Sackmann E (2005) Polymer-supported membranes as models of the cell surface. *Nature* 437: 656-63
- van Meer G, Voelker DR, Feigenson GW (2008) Membrane lipids: where they are and how they behave. *Nat Rev Mol Cell Biol* 9: 112-24
- van Pee K, Mulvihill E, Muller DJ, Yildiz O (2016) Unraveling the Pore-Forming Steps of Pneumolysin from *Streptococcus pneumoniae*. *Nano Lett* 16: 7915-7924
- Vigano E, Diamond CE, Spreafico R, Balachander A, Sobota RM, Mortellaro A (2015) Human caspase-4 and caspase-5 regulate the one-step non-canonical inflammasome activation in monocytes. *Nat Commun* 6: 8761

Thank you for submitting your manuscript to The EMBO Journal. Your study has now been re-reviewed by the referees and they very much appreciate the introduced changes. I am therefore very pleased to accept the manuscript for publication here. Before I can send you the formal acceptance letter there are just a few minor changes needed.

- Please respond to referees #2 last few comments either in the point-by-point response or in the text.

REFeree REPORTS.

Referee #1:

The authors did a very good job in revising their manuscript. My previous comments and criticisms have been sufficiently addressed. The manuscript shall be ready for publication now.

Referee #2:

The authors have addressed the reviewer's concerns adequately and the manuscript is ready for publication. Just a couple of minor questions:

- They explain why POPS, DOPE and POPI does not contribute to GSDMD pore formation. Can they also speculate why the POPS, DOPE and POPC system (with the same mol ratio) does? Does this mean that POPC somehow contributes to GSDMD pore formation as well, although to a less extend than PI(4,5)P2 ?

- In their reply to reviewer 1 they state that their AFM assay is sufficiently sensitive to follow the "prepore-to-pore transition", showing as an example an AFM study of PLY from a previous publication. In this case they clearly see the pore spanning completely the lipid bilayer (according to the pore thickness). I wonder if this is also the case for GSDMD, as apparently the height of the pore is around 2 nm (see for example FR1.1F and H = new Fig2). Could the authors comment on that?

Referee #3:

The authors have satisfactorily answered our questions and remarks. We have no further comments.

EMBOJ-2017-98321R "Mechanism of membrane pore formation by human Gasdermin-D"

Point-by-point response to the comments of reviewer #1

Reviewer #1: *The authors did a very good job in revising their manuscript. My previous comments and criticisms have been sufficiently addressed. The manuscript shall be ready for publication now.*

Authors: Thank you for your encouraging and constructive comments.

Point-by-point response to the comments of reviewer #2

Reviewer #2: *The authors have addressed the reviewer's concerns adequately and the manuscript is ready for publication. Just a couple of minor questions:*

Authors: Thank you for your encouraging and constructive comments, which guided us to revise our manuscript. Below please find in our point-by-point response to the two remaining questions of the referee.

Reviewer #2: - *They explain why POPS, DOPE and POPI does not contribute to GSDMD pore formation. Can they also speculate why the POPS, DOPE and POPC system (with the same mol ratio) does? Does this mean that POPC somehow contributes to GSDMD pore formation as well, although to a less extend than PI(4,5)P2 ?*

Authors: The reviewer asks whether POPC somehow contributes to GSDMD pore formation as well, although to a less extend than PI(4,5)P2. Indeed our data suggests that PI(4,5)P2 in the presence of POPS and DOPE (i.e. in SLMS were made from POPS, DOPE and PI(4,5)P2) supports more strongly GSDMD pore formation compared to POPC in the presence of POPS and DOPE (i.e. in SLMS made from POPS, DOPE and POPS) whereas POPI in the presence of POPS and DOPE (i.e. in SLMS made from POPS, DOPE and POPI) suppresses GSDMD pore formation.

Reviewer #2: - *In their reply to reviewer 1 they state that their AFM assay is sufficiently sensitive to follow the "prepore-to-pore transition", showing as an example an AFM study of PLY from a previous publication. In this case they clearly see the pore spanning completely the lipid bilayer (according to the pore thickness). I wonder if this is also the case for GSDMD, as apparently the height of the pore is around 2 nm (see for example FR1.1F and H = new Fig2). Could the authors comment on that?*

Authors: The reviewer asks whether we also observe that GSDMD pores spanning completely the lipid bilayer. In our AFM measurements, we observe that GSDMD pores span the supported lipid membranes at different depths. The depths measured range from 2–5 nm. Depending on the lipid composition the thickness of the supported lipid membrane ranges from \approx 3–5 nm. One could thus argue that in some of our measurements the AFM stylus (which has a radius of \approx 5–10 nm and an opening angle of \approx 45°) cannot fully penetrate through the pore to measure its full depth. This limitation caused by the dimension of the AFM stylus would particularly limit measuring the depth of GSDMD pores having smaller diameters. In fact, GSDMD pores show smaller diameters compared to PLY pores the reviewer refers to. One could also argue that the AFM stylus would be able to penetrate through the pore but that the pore is still filled with a monolayer of lipids (if this is possible in aqueous solution) or highly distorted lipid structures, which both could account for the reduced depth of the pore. Such distorted lipid structures, however, would hardly be able to seal the pore such as expected for intact lipid membranes.

Point-by-point response to the comments of reviewer #3

Reviewer #3: *The authors have satisfactorily answered our questions and remarks. We have no further comments.*

Authors: Thank you for your encouraging and constructive comments.

Corresponding Author Name: Daniel J. Müller

Manuscript Number: EMBOJ-2017-98321R